# FRUGAL: MEMORY-EFFICIENT OPTIMIZATION BY REDUCING STATE OVERHEAD FOR SCALABLE TRAINING

## ABSTRACT

With the increase in the number of parameters in large language models, the process of pre-training and fine-tuning increasingly demands larger volumes of GPU memory. A significant portion of this memory is typically consumed by the optimizer state. To overcome this challenge, recent approaches such as low-rank adaptation (LoRA (Hu et al., 2021)), low-rank gradient projection (GaLore (Zhao et al., 2024a)), and blockwise optimization (BAdam (Luo et al., 2024)) have been proposed. However, in all these algorithms, the *effective rank of the weight updates remains low-rank*, which can lead to a substantial loss of information from the gradient. This loss can be critically important, especially during the pre-training stage. In this paper, we introduce FRUGAL (**F**ull-**R**ank **U**pdates with **G**r**A**dient sp**L**itting), a new memory-efficient optimization framework. FRUGAL leverages gradient splitting to perform low-dimensional updates using advanced algorithms (such as Adam), while updates along the remaining directions are executed via state-free methods like SGD or signSGD (Bernstein et al., 2018). Our framework can be integrated with various low-rank update selection techniques, including GaLore and BAdam. We provide theoretical convergence guarantees for our framework when using SGDM for low-dimensional updates and SGD for state-free updates. Additionally, our method consistently outperforms concurrent approaches across various fixed memory budgets, achieving state-of-the-art results in pre-training and fine-tuning tasks while balancing memory efficiency and performance metrics.

## 1 INTRODUCTION

In recent years, Large Language Models (LLMs) such as GPT (OpenAI, 2023) and LLaMA-3 Dubey et al. (2024) have demonstrated remarkable performance across various disciplines (Brown, 2020b; Yang et al., 2024; Romera-Paredes et al., 2024). However, a critical factor in achieving these results is the size of these models (Hoffmann et al., 2022). A larger number of parameters not only increases computational cost but also significantly raises memory requirements. For instance, training an 8 billion parameter LLaMA model in a 16-bit format necessitates each parameter to occupy 2 bytes, resulting in 16GB for storing the parameters and an additional 16GB for gradients. Utilizing the Adam optimizer (Kingma, 2014), which is standard for pre-training and fine-tuning LLMs, adds a further 32GB of memory to store the $m$ and $v$ statistics, resulting in 64GB total amount of memory. Furthermore, to achieve higher-quality results, training in pure 16-bit format is often insufficient (Zamirai et al., 2020). This necessitates storing master weights and optimizer statistics in 32-bit format, leading to total memory demands that exceed the capacity of cutting-edge graphics cards, such as the A100-80GB.

Numerous research projects have been aimed at reducing these significant costs. These approaches include engineering solutions like gradient checkpointing Chen et al. (2016) and memory offloading (Rajbhandari et al., 2020), which do not change the training trajectory. There are also methods that adjust the training algorithm by decreasing the number of trainable parameters (Frankle & Carbin, 2018; Wang et al., 2023; Sreenivasan et al., 2022; Horváth et al., 2024) or their bit precision (Wortsman et al., 2023), as well as optimizer statistics (Dettmers et al., 2021; Shazeer & Stern, 2018; Zhang et al., 2024c). In this work, we concentrate on the latter category.

Parameter-Efficient Fine-Tuning (PEFT) methods, such as LoRA (Hu et al., 2021), Dora (Liu et al., 2024), and BitFit (Zaken et al., 2021) reduce memory costs by training a relatively small number of

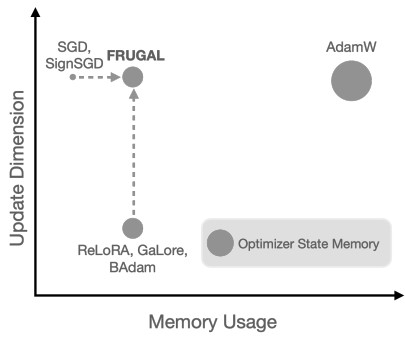

Figure 1: FRUGAL reduces memory usage by splitting gradient updates into low-dimensional updates with advanced optimizers (e.g., AdamW) and using state-free methods (e.g., SignSGD and SGD) for the rest.

**Algorithm 1** FRUGAL (State-Full, State-Free)

**Input:** model $f_\theta(\cdot)$ with $p$ parameters $\{\theta_i \in \mathbb{R}^{d_i}\}_{i=1}^p$, loss $\mathcal{L}$, gradient projectors $P_{k,i}$ for $i \in [p]$, number of steps $K$.

1: **for** $k = 1, 2, \ldots K$ **do**
2:      get data batch $(x, y)$
3:      compute $\ell \leftarrow \mathcal{L}(f_\theta(x), y)$           ▷ Forward
4:      **for** $g_i = \frac{\partial \ell}{\partial \theta_i}$ from Backward **do**
5:          $g_{\text{full},i} \leftarrow P_{k,i}(g_i),$        ▷ Project Grad
6:          $g_{\text{free, i}} \leftarrow g_i - P_{k,i}^{-1}(g_{\text{full},i})$     ▷ Residual
7:          $s_{\theta_i} \leftarrow [P_{k,i}(P_{k-1,i}^{-1}(s), \ s \in s_{\theta_i}]$▷ Project State s
8:          $u_{\text{full, i}} \leftarrow \texttt{State-Full.update}(\theta_i, g_{\text{full},i}, s_{\theta_i})$
9:          $u_{\text{free, i}} \leftarrow \texttt{State-Free.update}(\theta_i, g_{\text{free},i})$
10:        $\theta_i \leftarrow \theta_i + P_{k,i}^{-1}(u_{\text{full},i}) + u_{\text{free, i}}$
11:      **end for**
12: **end for**

parameters compared to the size of the original model, while the remaining modules are frozen. This approach has proven effective for the task of efficient fine-tuning of pre-trained language models. However, PEFT methods have a fundamental limitation: parameter updates always lie in a low-dimensional subspace $L$, which prevents the use of these methods for pre-training (Lialin et al., 2023) and may constrain their capabilities in fine-tuning (Zhang et al., 2024a).

Recent works, such as GaLore (Zhao et al., 2024a), ReLoRA (Lialin et al., 2023), BAdam (Luo et al., 2024) and BlockLLM (Ramesh et al., 2024), offer a solution to this problem. These methods enable higher-dimensional full-parameter learning by periodically changing the optimizable low-rank subspace $L$. However, even though these methods result in overall parameter changes that are high-dimensional, the updates in each step remain low-dimensional. The dimensionality of the frozen subspace $\dim M = \dim L^\perp$ significantly exceeds $\dim L$. The remaining information contained in the gradient is not utilized for parameter updates. Nevertheless, this information can still be leveraged to train the model.

We present the FRUGAL framework, designed to bridge this gap. Our approach stems from a crucial observation: although memory constraints prevent using optimizers with auxiliary optimizer state — such as Adam (Kingma, 2014) — in the remaining subspace $M$, one still can update $M$ using state-free optimization algorithms like Stochastic Gradient Descent (SGD) or signSGD (Bernstein et al., 2018). This solution allows for high-dimensional updates, which provides additional opportunities to explore the parameter space and improves convergence. We will further refer to subspaces $L$ and $M$ according to the types of optimizers used for their updates - **state-full** and **state-free**.

**Contributions.** We summarize the main contributions of our work as follows:

- We present a new memory-efficient optimization framework that combines the use of advanced optimization algorithms for the state-full subspace with state-free algorithms for the complementary subspace. The framework supports various types of state-full optimizers, state-free optimizers, and different methods for projecting the gradient onto the state-full subspace.

- We provide theoretical convergence guarantees for our framework. In the proof, we consider the case where SGDM acts as the state-full optimizer and SGD as the state-free optimizer, and we show that FRUGAL matches the best-known convergence rate in many scenarios.

- To verify the practical applicability of FRUGAL, we conduct extensive experiments in popular real-world scenarios.[1] In these experiments, we pre-train LLaMA-like models (up to 1B parameters) on the Colossal Clean Crawled Corpus (C4) dataset (Raffel et al., 2020) and fine-tune RoBERTa (Liu, 2019) on the GLUE benchmark (Wang, 2018). The results show that our method significantly outperforms previous memory-efficient algorithms while using the same memory budget.

- We demonstrate that only the Logits layer in transformer-like models requires advanced optimizers like Adam, while other modules (including Embeddings and RMSNorms) can use simpler methods

---

[1]The code is available at `https://anonymous.4open.science/r/frugal-666D`.

like signSGD without significant performance loss. This opens new possibilities for memory-efficient training and provides crucial insights into the learning dynamics of Transformers.

## 2 RELATED WORK

**Memory-efficient full-parameter learning.** Recent research has focused on reducing the memory footprint of LLMs by decreasing the size of optimizer states while maintaining their performance. Low-rank adaptation methods, such as LoRA (Hu et al., 2021), inject trainable rank decomposition matrices into each layer of the model, reducing memory requirements by optimizing only a few learnable adapters. ReLora (Lialin et al., 2023) builds upon this by merging low-rank adaptations into the main model weights during training, potentially increasing the total rank of the update. BAdam (Luo et al., 2024) leverages Block Coordinate Descent for full-parameter training by switching active blocks during fine-tuning. MicroAdam (Modoranu et al., 2024) compresses gradient information before feeding it into the optimizer state, significantly reducing memory footprint while enabling full parameter learning through error feedback mechanisms. GaLore (Zhao et al., 2024a) maintains full parameter learning by projecting gradients onto a low-rank subspace using truncated SVD decomposition, storing optimizer states in this reduced space. Notably, GaLore achieves good results in pre-training, with performance close to that of Adam. However, while these methods effectively reduce memory overhead, they all perform *low-rank updates at each iteration*. In contrast, our approach utilizes all available gradient information to perform *full-dimensional updates at each optimizer step*, offering a novel perspective on memory-efficient optimization for LLMs.

**Other memory-efficient optimization.** Several other methods have been proposed to reduce the memory footprint of optimizers. AdaFactor (Shazeer & Stern, 2018) attempts to mimic Adam's behavior while reducing memory usage through factorization of the variance matrix $v$. Adam-mini (Zhang et al., 2024c) further reduces memory by storing only one value $v$ per block. Dettmers et al. (2021) and Li et al. (2024) decrease memory footprint by quantizing optimizer states to the lower-precision representations. Lv et al. (2023) proposed to reduce weight gradient memory by fusing the backward operation with the optimizer update. Notably, these approaches are orthogonal to our method FRUGAL and *can be combined with it* for further memory efficiency.

**Block Coordinate Descent.** Block Coordinate Descent (BCD) is a well-established optimization method with a rich history in mathematical optimization (Ortega & Rheinboldt, 2000; Tseng, 2001; Richtárik & Takáč, 2014; 2015b; Richtárik & Takác, 2016; Takáč et al., 2013; Richtárik & Takáč, 2015a). In recent years, a specific instance of BCD, known as *layer-wise learning,* has been applied to deep learning. Notable examples include (Luo et al., 2024; Pan et al., 2024), which leverage this approach for LLM fine-tuning. To the best of our knowledge, our work presents **the first theoretical analysis** of an extended BCD framework (Section 5) where the *remaining coordinates are also updated using a different algorithm*. This novel approach extends traditional BCD techniques, opening new avenues for full model optimization in deep learning.

**Sign-based methods for training language models.** Since its introduction, Adam has become the de facto primary optimization algorithm, demonstrating superior practical results compared to SGD-based algorithms across various deep learning tasks. This difference is particularly noticeable when training Transformers on language tasks. While Zhang et al. (2020) hypothesized that Adam outperforms SGD in this setup due to *the heavy-tailed distribution of sampling-induced errors,*

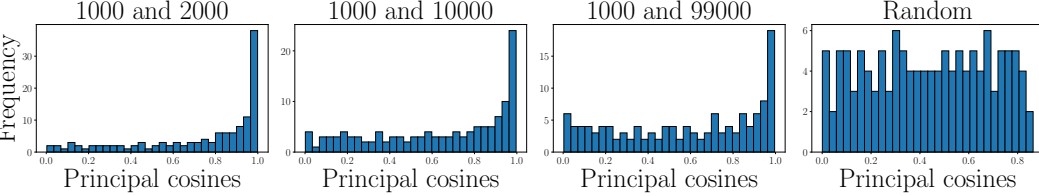

Figure 2: Histograms of principal angle cosines. The first three are taken between $\boldsymbol{P}_t$ and $\boldsymbol{P}_{t'}$ from different iterations $t$ and $t'$. $\boldsymbol{P}$ is obtained from the truncated SVD decomposition of the gradient $\boldsymbol{G}$ of the Key projection from the 5th layer. The last histogram is taken between two random semi-orthogonal projections $\boldsymbol{R}$ and $\boldsymbol{R}'$ for comparison.

Table 1: Comparison of different projection and state-free subspace optimization strategies on pre-training LLaMA-130M on C4 with Adam as the state-full algorithm.

| Projection type | Optimizes state-free subspace | Validation perplexity after iterations ↓ | | | | |
|---|---|---|---|---|---|---|
| | | 4k | 20k | 40k | 100k | 200k |
| SVD | No | **39.75** | 26.35 | 24.38 | 22.30 | 21.11 |
| Random | No | 42.31 | **25.99** | **23.55** | **21.33** | **20.01** |
| Random | Yes | 37.26 | 23.46 | 21.53 | 19.66 | 18.64 |
| SVD | Yes | **33.96** | **22.54** | **21.01** | **19.30** | **18.35** |
| RandK | Yes | 36.38 | 23.02 | 21.25 | 19.70 | 18.63 |
| Blockwise | Yes | 37.20 | 23.34 | 21.42 | 19.59 | 18.60 |
| Adam | | 33.95 | 21.90 | 20.56 | 18.97 | 18.13 |

Kunstner et al. (2023) demonstrated that this superiority persists even in full-batch training. They proposed a new hypothesis suggesting that Adam's key success factor is related to *its similarity to signSGD* (Balles & Hennig, 2018; Balles et al., 2020), and both Kunstner et al. (2023) and Zhao et al. (2024b) showed that signed descent with momentum reduces the performance gap with Adam. In contrast, to the best of our knowledge, *we are the first to train the majority of language model's parameters using signSGD without momentum*, achieving minimal loss in quality. This approach further demonstrates the effectiveness of sign-based methods for LLM training, paving the way for more efficient and scalable optimization strategies in deep learning.

## 3 EMPIRICAL ANALYSIS AND MOTIVATION

In this section, we present empirical evidence that motivates our approach. First, we show that access to the whole parameter space is crucial during training. Then, we show how utilizing full-rank updates can significantly improve model performance.

### 3.1 THE IMPORTANCE OF EXPLORING THE ENTIRE SPACE DURING THE TRAINING PROCESS

In recent work, Zhao et al. (2024a) proposed GaLore, an optimization method based on projecting the gradient matrix $G$ of each Linear layer[2] onto a low-dimensional subspace. To obtain the projection matrix $P$, they use SVD decomposition of $G_t$, which is recomputed with frequency $T$. The vectors or rows of $G$ are projected onto the first $r$ left or right singular vectors, respectively. This approach has theoretical foundations: the first $r$ singular vectors correspond to the first $r$ singular values and, therefore, should better utilize information from the spectrum of $G$.

The authors pointed out that calculating the SVD decomposition results in extra computational overhead, which can be as much as a 10% increase as the hidden size of the model grows. To minimize this cost and examine the significance of using SVD decomposition, one may wonder about the possibility of employing a random semi-orthogonal projection matrix $R$ instead of projecting onto the first $r$ singular columns with $P$. Surprisingly, while SVD decomposition provides better initial performance, random projection proves superior in long-term training, yielding significant improvements. As an illustration, we took the pre-training[3] of a 130M model with LLaMA-like architecture on the C4 dataset (Raffel et al., 2020). The results are presented in the first part of Table 1 (`Optimizes state-free subspace = No`), where we compare SVD and Random projections. The ranks of both projections $P$ and $R$ are equal to 192.

To investigate this phenomenon, we pre-trained the LLaMA-60M model and collected gradients $G_t$ from different iterations $t$ for examination. Following the setup from GaLore (Zhao et al., 2024a), we computed SVD decompositions and extracted projections $P_t$ with a rank of 128. We evaluated the similarity of the projection matrices by calculating the principal angles between different projections $P_t$ at different steps. Similarly to the observations in Q-Galore (Zhang et al., 2024d), we found that these projections show minimal change during the training period; see Figure 2.

---

[2]Since Linear layers contain most parameters and require most memory, we primarily focus on them.

[3]See Section 6.1 for a detailed description and discussion on the experimental setup.

Here, we take the projection matrix corresponding to the 5-th layer and plot histograms of the cosine of the principal angles between pairs $P_t$ and $P_{t'}$ from different iterations. For comparison, we also include the random projections on the right. As can be seen, the distributions of cosines differ significantly for the $P_t$ and for the random projections. While $R_t$ feature no angles with cosines higher than $0.9$, the top $57$ cosines for $P_t$ surpass $0.9$, even for gradients $1000$ steps apart.

This leads to the conclusion that although SVD decomposition generally better captures the information contained in the $G_t$, the original GaLore algorithm updates weights only in a small subspace. We hypothesize that training with random projections yields superior results due to the more extensive investigation of the optimizable space during the training process. *This finding indicates that to achieve better convergence, it is important to seek out optimization algorithms that explore the entire space during the training process.*

### 3.2 Advantage of the Full-Rank Updates

The insight from the Section 3.1 suggests that the training of language models performs significantly better when the entire parameter space is utilized during the training process. Given the importance of updating parameters in all directions, this poses the question: *Is it optimal to use low-rank updates, as employed by methods such as GaLore (Zhao et al., 2024a), ReLoRA (Lialin et al., 2023), and BAdam (Luo et al., 2024)?* Using low-rank updates means the effective rank of the update is significantly smaller than the full dimensionality of the parameter space, inevitably leading to a loss of valuable information contained in the gradient.

However, the method to leverage the full-rank gradient for updating parameters is not readily obvious. Using algorithms like Adam (Kingma, 2014) is not an option due to the memory overhead they introduce, which is precisely what we aim to avoid. An alternative approach is to use state-free optimizers such as SGD or signSGD (Bernstein et al., 2018). Unfortunately, SGD have been shown to be ineffective for training transformer models, as shown in Zhang et al. (2020); Pan & Li (2023).

Nevertheless, a recent study Zhao et al. (2024b) suggests a promising methodology: while SGDM doesn't generally work well with transformers, using SGDM for the majority of parameters and Adam for a selected subset can lead to effective training. This raises the question: could a hybrid approach using SGD or signSGD instead of SGDM be viable? If the key subset of parameters is handled by advanced algorithms, can the other parameters be trained effectively with state-free optimizers?

To address this question, we conducted an experiment on LLaMA-130m, where we utilized the Adam (Kingma, 2014) for state-full parameters and signSGD (Bernstein et al., 2018) for state-free parameters. A detailed description of the experimental setup can be found at Appendix A.1. Once again we used Random projection and highlighted the result in the second part of Table 1 (Optimizes state-free subspace = Yes). Full-rank updates significantly enhance performance, approaching the efficiency of the memory-intensive Adam optimizer, which serves as an upper bound in terms of performance. *These findings underscore the potential of state-free algorithms for updating a substantial portion of the parameter space, paving the way for efficient, scalable optimization methods that deliver high performance without the significant memory costs traditionally associated with state-of-the-art optimizers.*

## 4 FRUGAL: Full-Rank Updates with GrAdient spLitting

**General framework.** The setup outlined at the conclusion of the Section 3.2 results in a general framework for memory-efficient optimization. It operates as follows: the entire space is partitioned into *state-full* and *state-free* subspaces. The state-full subspace is updated using advanced algorithms, while the state-free subspace is updated using a state-free method. After a certain number of steps, the state-full subspace is changed to better explore the optimization space. A formal description of the final algorithm is presented in Algorithm 1. We note that this framework allows for variation not only in the `State-Full` optimizer but also in the choice of projection and `State-Free` optimizer.

However, determining the optimal state-free optimizer and the projection method onto the state-full subspace is not readily apparent. In this section, we strive to find the optimal configuration.

**State-free optimizer.** We conducted a preliminary experiment updating all parameters using state-free algorithms to choose between SGD and signSGD (Bernstein et al., 2018). Table 8 presents these

---

**Algorithm 2** FRUGAL (SGDM, SGD)

---

**Input:** momentum weight $\beta \in [0, 1)$, initialization $x^1 \in \mathbb{R}^d$ and $m^0 = 0$, step sizes $\{\alpha^k > 0\}_{k=1}^K$, momentum set $J_k \subset [d]$ for $k = 1, 2, \ldots$.

1: **for** $k = 1, 2, \ldots$ **do**
2:      Compute stochastic gradient $\tilde{g}^k \leftarrow \nabla f_{\zeta^k}(x^k)$;
3:      Update momentum vector $\tilde{m}_j^k \leftarrow (1 - \beta)\tilde{g}_j^k + \beta \begin{cases} \tilde{m}_j^{k-1} & \text{if } j \in J_k, \\ 0 & \text{otherwise}; \end{cases}$
4:      Compute update vector $\tilde{u}_j^k \leftarrow \begin{cases} \tilde{m}_j^k & \text{if } j \in J_k, \\ \tilde{g}_j^k & \text{otherwise}; \end{cases}$
5:      Update iterate $x^{k+1} \leftarrow x^k - \alpha^k \tilde{u}^k$;
6: **end for**

---

results. After testing various learning rates, we found that signSGD consistently outperforms SGD, leading us to favor signSGD. We attribute this performance to the similarities between signSGD and Adam (Kingma, 2014), as noted in Balles & Hennig (2018); Balles et al. (2020); Kunstner et al. (2023). Additionally, signSGD produces updates of similar magnitude to those generated by Adam, which simplifies the calibration of the learning rate for state-free parameters.

**Projection type.** When selecting a projection method, it is crucial to strike a balance between quality and memory efficiency. When using SVD decomposition for projection matrices, as in GaLore (Zhao et al., 2024a), the method better preserves the information embedded in the gradient but requires additional memory for storing projection matrices and computational resources for performing the SVD. To reduce computational demands, one could employ random coordinate projection denoted as RandK, but this requires additional memory or recomputation[4]. A more structured alternative is to select not random entries but entire random columns or rows. The most aggressive approach follows the method from BAdam, wherein an entire block is chosen as the state-full subspace.

The performance results obtained with all these variants are presented in the second part of Table 1. SVD outperforms both RandK and Block projections, demonstrating comparable performance. The superior performance of SVD projection can be explained by its ability to extract the principal information from the gradient. Nonetheless, a downside is the increased compute and memory demand from SVD. Therefore, we opt for the blockwise selection, as it is the most memory-efficient — requiring only the storage of active block indices.

In our experiments, we use a specific variant with Adam as the State-Full optimizer and signSGD as the State-Free optimizer. We primarily employ blockwise projection but switch to column-wise projection when the number of parameters in any single block exceeds memory budget, as detailed in Section 6.2. In addition, PyTorch-like pseudocode of our framework is presented in Appendix G.

For Line 7, state projection, in Algorithm 1, we note that if the projection does not change, i.e., $P_{k,i} = P_{k-1,i}$, then $P_{k,i}(P_{k-1,i}^{-1}(s)) = s$. Thus, we only need to project states when the projection changes from one round to another. However, our preliminary experiments with RandK selection showed that resetting states performs comparably to projection. Therefore, we could replace this projection with state resetting when the projection changes, which also aligns with blockwise subspace selection. However, either resetting or projecting states is important since we want projected gradients and optimizer states to reside in the same space. For instance, GaLore ignores this step, which leads to degraded performance when projections are updated frequently; see Appendix C for details.

## 5 THEORETICAL RESULTS

For the theoretical analysis, we consider the case where the `State-Free` optimizer is SGD and the `State-Full` optimizer is SGD with momentum (SGDM). For the projection, we use coordinate-wise projection. This special case of FRUGAL is provided in Algorithm 2. We minimize the objective

$$\min_{x \in \mathbb{R}^d} \left\{ f(x) := \mathbb{E}_{\zeta^k}[f_{\zeta^k}(x)] \right\}, \tag{1}$$

where we access $f$ via a stochastic oracle that takes $x$ as input and returns $(f_{\zeta^k}(x), \nabla f_{\zeta^k}(x))$.

---

[4]See Appendix B for discussion on the memory requirements for different projection methods.

## 5.1 NOTATION AND PRELIMINARIES

We use $\|\cdot\|$ for the vector $\ell_2$-norm, and $\langle\cdot,\cdot\rangle$ stands for the dot product. Let $g^k$ denote the full gradient of $f$ at $x^k$, i.e., $g^k := \nabla f(x^k)$, $\tilde{g}^k$ denote the stochastic gradient $\tilde{g}^k = \nabla f_{\zeta^k}(x^k)$ for random sample $\zeta^k$, and $f^* := \min_{x\in\mathbb{R}^d} f(x)$. We use subscript $j$ to denote the $j$-th coordinate. We call a function $L$-smooth if it is continuously differentiable and its gradient is Lipschitz continuous:

$$\|\nabla f(x) - \nabla f(y)\| \leq L\|x - y\|. \tag{2}$$

**Assumption 1.** *We make the following assumptions, which are standard in non-convex stochastic optimization; see (Liu et al., 2020).*

1. **Smoothness:** *The objective $f(x)$ in equation 1 is $L$-smooth (eq. (2)).*

2. **Unbiasedness:** *At each iteration $k$, $\tilde{g}^k$ satisfies $\mathbb{E}_{\zeta^k}[\tilde{g}^k] = g^k$.*

3. **Independent samples:** *The random samples $\{\zeta^k\}_{k=1}^{\infty}$ are independent.*

4. **Bounded variance:** *The variance of $\tilde{g}_j^k$ with respect to $\zeta^k$ satisfies $\mathrm{Var}_{\zeta^k}(\tilde{g}_j^k) = \mathbb{E}_{\zeta^k}[\|\tilde{g}_j^k - g_j^k\|^2] \leq \sigma_j^2$ for some $\sigma_j^2 > 0$. We denote $\sigma^2 = \sum_{j=1}^{d}\sigma_j^2$.*

Finally, we define the probability that index $j \in J_k$ is selected, conditioned on the prior iteration $k-1$, as $p_j^k := \mathrm{Pr}_{k-1}[j \in J_k]$. Other useful quantities are $p_{\max}^k := \max_{j\in[d]}\{p_j^k\}$ and $p_{\min}^k := \min_{j\in[d]}\{p_j^k\}$.

## 5.2 CONVERGENCE OF ALGORITHM 2

Below, we present the main convergence theorem.

**Theorem 1.** *Let Assumption 1 hold and $\alpha^k = \alpha \leq \frac{1-\beta}{L(4-\beta+\beta^2)}$. Then, the iterates of Algorithm 2 satisfy*

$$\frac{1}{k}\sum_{i=1}^{k}\mathbb{E}[\|g^i\|^2] = \mathcal{O}\left(\frac{f(x^1) - f^*}{k\alpha} + L\alpha\sigma^2\left(1 + \frac{\hat{p}_{\max}^k(1-\bar{p}_{\min}^k)\beta}{(1-\beta)}\right)\right), \tag{3}$$

*where $\bar{p}_{\min}^k = \frac{1}{k}\sum_{i=1}^{k}\bar{p}_{\min}^i$ and $\hat{p}_{\max}^k = \max_{i\in[k]}\{p_{\max}^i\}$.*

The proof is deferred to Appendix E. Let us analyze the obtained result. Firstly, if $J_k = [d]$ or $J_k = \emptyset$, Algorithm 2 becomes SGDM and SGD, respectively. In this case, we have $\bar{p}_{\min}^k = 1$ for SGDM and $\hat{p}_{\max}^k = 0$ for SGD. Therefore, the resulting rate is $\mathcal{O}\left(1/k\alpha + L\alpha\sigma^2\right)$, which recovers the best-known rate for both SGD and SGDM under these assumptions Liu et al. (2020). Furthermore, if at each step each coordinate is sampled independently with probability $p$, we have $\bar{p}_{\min}^k = \hat{p}_{\max}^k = p$. Therefore, we recover the same rate if $p = \mathcal{O}(1-\beta)$ or $p = \mathcal{O}(\beta)$. Finally, in the worst case (e.g., $J_k$ is deterministic and $0 < |J_k| < d$), we have $\bar{p}_{\min}^k = 0$ and $\hat{p}_{\max}^k = 1$. Thus, the rate becomes $\mathcal{O}\left(1/k\alpha + L\alpha\sigma^2/1-\beta\right)$, which is worse by a factor of $1/1-\beta$. However, this is expected since the bias from momentum is not outweighed by the variance reduction effect, as only the coordinates with momentum enjoy reduced variance; see Lemmas 1 and 2 in the appendix for details.

# 6 EXPERIMENTS

This section presents the main experimental results of the paper. To evaluate the performance of FRUGAL, we conducted experiments both on the pre-training and fine-tuning of language models.

## 6.1 PRE-TRAINING EXPERIMENTS

**Setup.** The core setup for pre-training is taken from the Zhao et al. (2024a). We utilize LLaMA-based (Touvron et al., 2023a) model architectures with up to 1B parameters and train them on the Colossal Clean Crawled Corpus (C4) dataset (Raffel et al., 2020). The C4 dataset is intended for pre-training, making this setup a good approximation of real-world applications. A detailed description of the setup can be found in Appendix A.1. However, we made several modifications that we would like to discuss in detail below.

Table 2: Comparison of validation perplexity and memory estimation for various optimization methods across LLaMA model scales trained on C4. We also indicate the additional memory overhead introduced by the optimization algorithm. The values are calculated assuming that each float value occupies 4 bytes (float32). $\rho$ denotes the proportion of the Linear layer parameters in the state-full subspace. Note that Embeddings, RMSNorms, and Logits are always trained with Adam.

|  | 60M | 130M | 350M | 1B |
|---|---|---|---|---|
| Adam | 22.73 (0.43G) | 18.13 (1.00G) | 14.43 (2.74G) | 12.02 (9.98G) |
| GaLore, $\rho = 0.25$ | 25.68 (0.30G) | 21.11 (0.54G) | 16.88 (1.10G) | 13.69 (3.41G) |
| BAdam, $\rho = 0.25$ | 24.86 (0.29G) | 20.34 (0.52G) | 16.41 (1.05G) | 13.75 (3.23G) |
| FRUGAL, $\rho = 0.25$ | **23.59** (0.29G) | **18.60** (0.52G) | **14.79** (1.05G) | **12.32** (3.23G) |
| FRUGAL, $\rho = 0.0$ | 24.06 (0.24G) | 18.90 (0.37G) | 15.03 (0.49G) | 12.63 (0.98G) |
| Training tokens | 20B | 20B | 24B | 30B |
| Number of iterations | 200k | 200k | 240k | 300k |

- **Training Duration.** The training approach in Zhao et al. (2024a) aligns with the empirical rule from scaling laws (Hoffmann et al., 2022), which suggests using approximately 20 times the model size in tokens for training. However, this number of tokens is far from achieving convergence. In practice, models are typically trained for significantly longer periods (Touvron et al., 2023b; Zhang et al., 2024b). One reason for this discrepancy is that the original scaling laws do not account for the inference of the model after training. Adjustments to scaling laws considering this parameter are discussed, for example, in (Sardana & Frankle, 2023). For our experiments we chose 200k steps for the 60M and 130M models, 240k for 350M model and 300k for the 1B model.

- **Learning Rate.** The authors of GaLore suggested using different learning rates for fixed unprojectable parameters (Embeddings, RMSNorms (Zhang & Sennrich, 2019), Logits) and the remaining projectable parameters (attention and FFN weights modules weights). However, introducing additional hyperparameters complicates the use of the algorithm. Since both sets of parameters are state-full and trained using the same optimization algorithm, we always used the same learning rate for them in FRUGAL and BAdam. For GaLore learning rate see Section 6.1.

- **Mixed Precision instead of the pure bfloat16 training.** Pure 16-bit training has been shown to potentially compromise model convergence and accuracy (Zamirai et al., 2020). This degradation stems from storing both the model weights and optimizer statistics in reduced precision formats such as float16 or bfloat16. However, these formats often lack sufficient precision in representing floating-point numbers. Consequently, mixed precision training has become a more common approach for training language models (Le Scao et al., 2023; Almazrouei et al., 2023)). While training in pure 16-bit format is also possible, stochastic rounding (Gupta et al., 2015; Zamirai et al., 2020) is often employed to mitigate the aforementioned issue. Given that the goal of this research is to identify the optimal optimization algorithm, we deemed it more appropriate to compare optimizers in a transparent and stable setup that does not require auxiliary tricks. Hence, we primarily used Mixed Precision training for its illustrative value in understanding each method's potential. However, for completeness, we also conducted experiments in pure bfloat16 format, detailed in our ablation study Section 6.1.2.

### 6.1.1 COMPARISON TO EXISTING MEMORY-EFFICIENT ALGORITHMS

To begin, we present the results of comparing FRUGAL with existing memory-efficient methods across four sizes of LLaMA-based architectures: 60M, 130M, 350M, and 1B[5].

**Baselines.** We use the following methods as baselines for our approach:

- **Full-rank Training.** Training using memory-inefficient Adam. Weights, gradients, and statistics are stored and computed for all parameters. This serves as an upper bound for model performance.

- **GaLore.** Zhao et al. (2024a) proposed GaLore, a memory-efficient optimization algorithm that uses a low-rank projection of gradient matrices $G$. Every $T$ steps, the current gradient matrix $G_t$ is

---

[5]See preliminary experimental results with LLaMA 7B in Appendix D

Table 3: Perplexity of LLaMA-130M models pre-trained on C4 for 100k iterations (10B tokens). The leftmost column indicates the modules moved to the state-free set and trained using signSGD. The results show that **Logits**, unlike Embeddings and RMSNorms, are exceptionally responsive to the choice of optimization algorithm from Adam to signSGD.

| State-free modules | Perplexity $\downarrow$ |
|---|---|
| Linear (corresponds to the `FRUGAL` with $\rho = 0.0$ from Table 2) | 20.02 |
| Linear, *RMSNorms* | 20.07 |
| Linear, *Embeddings* | 20.48 |
| Linear, *Embeddings, RMSNorms* | 20.55 |
| Linear, **Logits** | 34.66 |

used to compute the projection matrix $P$ via SVD decomposition. The gradient is then projected onto the low-rank space, where the optimization step is performed. Subsequently, the resulting low-rank update is projected back into the full-rank space and added to the weights $W$.

- **BAdam.** Luo et al. (2024) proposed a block coordinate descent (BCD)-type optimization method termed BAdam. The parameters are divided into blocks, which are then updated one by one using Adam. Similar to GaLore, the optimized block is updated every $T$ steps. Although this method was initially proposed only for fine-tuning, it is the closest method to our `FRUGAL`. Unlike BAdam, in our algorithm, state-free blocks are not frozen but are updated using signSGD.

- **Other Algorithms.** Among other relevant methods, ReLoRA (Lialin et al., 2023) and MicroAdam (Modoranu et al., 2024) can also be highlighted. However, we did not include them for comparison in this paper for the following reasons: 1. ReLoRA was evaluated in (Zhao et al., 2024a), where it significantly underperformed compared to GaLore with the same memory budget. 2. MicroAdam. Its current implementation only supports bfloat16 master weights, whereas our main experiments conducted with mixed precision.

**Main results.** The results of our experiments are presented in Table 2, which includes both validation perplexity and memory footprint estimations for each method. We compared all memory-efficient methods under the same memory budget with a density $\rho = 0.25$. Here, $\rho$ refers to the proportion of Linear layer parameters in the state-full subspace. Similarly to GaLore, non-Linear modules (Embeddings, RMSNorms, Logits) are optimized with Adam. See Appendix A.1 for details.

We conducted a grid search to determine the optimal learning rate for Adam, which we then applied uniformly to `FRUGAL` and BAdam (Luo et al., 2024). For GaLore (Zhao et al., 2024a), we found that using this same learning rate produced better results than the rate originally suggested in their paper. This discrepancy might be attributed to our experiments involving a significantly larger number of training steps than those for which GaLore's original learning rate was optimized.

Table 2 demonstrates that `FRUGAL` significantly outperforms the memory-efficient baselines across all model sizes with the same memory budget, coming close to the performance of Adam.

**Zero-density training.** Table 2 also reveals a surprising result: `FRUGAL` with $\rho = 0.0$ outperforms both GaLore and BAdam, even when these competing methods use a higher density of $\rho = 0.25$. Essentially, for `FRUGAL` with $\rho = 0.0$, the parameters are divided into two parts — a state-full part consisting of the Embeddings, RMSNorms, and Logits, and a state-free part consisting of all other parameters. This division remains fixed throughout the training. We conducted additional experiments to determine the maximum subset of parameters that can be trained with a state-free optimizer without significant quality degradation. We systematically moved different combinations of the Embeddings, RMSNorms, and Logits from the state-full to the state-free set and observed the results during the training of LLaMA-130M. Table 3 reveals that the Logits demonstrates a dramatically higher sensitivity, with changes to its optimizer resulting in severe performance degradation. This finding aligns with results from (Zhao et al., 2024b), where the authors demonstrated that most parameters can be trained using SGDM, but the Logits require training with Adam.

### 6.1.2 ABLATION STUDY

We also conducted additional experiments to verify the robustness of our framework to various hyperparameters. Firstly, an ablation study on the state-full subspace update frequency $T$ in Table 10

shows that the performance keeps improving up to $T = 200$. We note that, unlike in Zhao et al. (2024a), the perplexity does not significantly decrease even when reducing the update frequency to $T = 10$ ($\sim 0.2$ drop vs. $\sim 4$. drop for GaLore). A detailed explanation for this result can be found in Appendix C. Second, when using other schedulers, the performance gap between FRUGAL and baselines remains consistent, as shown in Tables 5 and 6. Then, the results of training in pure bfloat16 are presented in Table 7, demonstrating consistency with our main experiments in Table 2, i.e., FRUGAL significantly outperforms the baselines across these variations. We also conducted experiments to show how perplexity changes with varying $\rho$, and the results are presented in Table 11. Finally, we conducted an experiment to compare different strategies for selecting state-full blocks during training. The results in Table 9 show that there is no significant difference between random and structured block selection.

## 6.2 Fine-tuning experiments

Table 4: Evaluating FRUGAL for memory-efficient fine-tuning RoBERTa-Base on GLUE benchmark. Results represent the mean and standard deviation across 3 independent runs. Upper $\uparrow$ is better.

| Method | Modules | Rank | CoLA | STS-B | MRPC | RTE | SST2 | MNLI | QNLI | QQP | Avg |
|---|---|---|---|---|---|---|---|---|---|---|---|
| Full-parameter | — | — | 63.6 | **91.2** | **90.2** | 78.7 | 94.8 | **87.6** | 92.8 | **91.9** | 86.4 |
| LoRA | QV | 8 | $63.8_{\pm.6}$ | $90.9_{\pm.1}$ | $89.1_{\pm.4}$ | $79.2_{\pm1.1}$ | $94.8_{\pm.2}$ | $\mathbf{87.6}_{\pm.2}$ | $93.1_{\pm.1}$ | $90.6_{\pm.0}$ | 86.1 |
| GaLore | All | 8 | $60.0_{\pm.2}$ | $90.8_{\pm.1}$ | $89.0_{\pm.7}$ | $79.7_{\pm.9}$ | $\mathbf{94.9}_{\pm.5}$ | $\mathbf{87.6}_{\pm.1}$ | $\mathbf{93.3}_{\pm.1}$ | $91.1_{\pm.1}$ | 85.8 |
| GaLore | QV | 8 | $56.1_{\pm.8}$ | $90.8_{\pm.2}$ | $88.1_{\pm.3}$ | $74.7_{\pm1.9}$ | $94.3_{\pm.1}$ | $86.6_{\pm.1}$ | $92.6_{\pm.1}$ | $89.4_{\pm.1}$ | 84.1 |
| FRUGAL | QV | 8 | $64.5_{\pm.7}$ | $91.1_{\pm.1}$ | $89.2_{\pm.3}$ | $\mathbf{82.4}_{\pm.9}$ | $94.8_{\pm.2}$ | $87.4_{\pm.1}$ | $92.8_{\pm.1}$ | $91.4_{\pm.1}$ | **86.7** |
| FRUGAL | None | 0 | $\mathbf{64.8}_{\pm.5}$ | $91.1_{\pm.1}$ | $89.1_{\pm.3}$ | $81.6_{\pm.6}$ | $\mathbf{94.9}_{\pm.2}$ | $87.3_{\pm.1}$ | $92.8_{\pm.1}$ | $91.3_{\pm.1}$ | 86.6 |

We evaluated the performance of our framework in the context of memory-efficient fine-tuning using the GLUE benchmark (Wang, 2018), a widely-used collection of tasks for evaluating language models. Following the approach from Zhao et al. (2024a), we fine-tuned RoBERTa-base (Liu, 2019) using LoRA (Hu et al., 2021) and GaLore as baselines for comparison. We adhered to the setup described in LoRA, where low-rank updates of rank 8 were applied only to the Q and V matrices. For a detailed description of the experimental setup, see Appendix A.2.

However, this comparison required a minor modification to FRUGAL compared to the pre-training phase. Instead of selecting active parameters blockwise, we opted for columnwise selection in each matrix. This adjustment was necessary to ensure a fair comparison within a similar memory budget, as the number of trainable parameters in LoRA with rank 8 is approximately 2.5 times fewer than the number of parameters in any RoBERTa matrix. This transition from blockwise to columnwise selection allowed us to maintain comparable memory usage across methods. For the same reason, we did not include comparisons with BAdam (Luo et al., 2024) in this setup.

The results are presented in Table 4. Since the LoRA setup adds trainable adapters only to the Q and V matrices, while the GaLore code uses all modules as projectable parameters, we conducted experiments in both setups. The Full-parameter results are taken from the prior works. The results demonstrate that FRUGAL significantly outperforms GaLore and shows comparable results to LoRA.

As in Section 6.1.1, we conducted additional experiments with FRUGAL using $\rho = 0.0$. In this setup, only the classification head is trained using Adam, while the embedding parameters remain frozen, and the remaining parameters are trained using signSGD. The results demonstrate that this training approach barely compromises performance compared to FRUGAL with rank 8, and still outperforms GaLore. Similar to our findings in Section 6.1.1, we observe that the classification head parameters are particularly sensitive to the choice of optimizer, which can be seen in Table 13 where the model's performance significantly deteriorates when using signSGD for classification head optimization.

## 7 Conclusion

In this work, we introduce a new memory-efficient optimization framework, FRUGAL. Within this framework, the optimization space is divided into two subspaces: the first is updated using a state-full algorithm such as Adam, while the second is updated using a state-free algorithm such as signSGD. We prove theoretical convergence guarantees for our framework with SGDM serving as the state-full algorithm and SGD as the state-free algorithm. In experiments involving pre-training and fine-tuning of language models, FRUGAL outperforms other approaches while using the same or smaller memory.

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

# A  EXPERIMENTAL SETUPS

This section describes the main setups used in the experiments and presents additional experiments.

To begin, we introduce the hyperparameter density $\rho$. This hyperparameter represents the fraction of the total space in Linear layers that is updated with a stateful optimizer. For GaLore, this parameter is equal to $\rho = r/h$, where $r$ is the projection rank, and $h$ is the hidden size of the model. For RandK projection, this parameter can be expressed as $1 - s$, where $s$ means sparsity. For BAdam and FRUGAL with the blockwise update, this parameter denotes the ratio of the number of active blocks $a_{\text{block}}$ to the total number of blocks $p$, i.e., $\rho = a_{\text{block}}/p$. When using FRUGAL with the column-wise update, as in Section 6.2, $\rho$ is equal to the ratio of the number of active columns $a_{\text{column}}$ to their total number $h$, i.e., $\rho = a_{\text{column}}/h$.

## A.1  PRE-TRAINING SETUP

We adopt a LLaMA-based architecture with RMSNorm and SwiGLU (Wang, 2018) activations on the C4 dataset. Following Zhao et al. (2024a), we trained using a batch size of 512 sequences, sequence length of 256, weight decay of 0, and no gradient clipping. We used T5 tokenizer, since it also was trained on C4 with dictionary size equal to 32k. The update frequency $T$ is set to 200.

Since, unlike GaLore, we consider not only matrix projections, we decided to generalize the concept of rank $r$. Instead, we use density $\rho$, which represents the proportion of Linear layer parameters in the state-full subspace. Thus, for SVD-like projection as in GaLore, the density equals $\rho = r/h$, where $h$ denotes the hidden dimension of the model. We also should point out that similarly to Zhao et al. (2024a), we keep Embeddings, RMSNorms (Zhang & Sennrich, 2019), and Logits in the state-full subspace throughout the training and don't reset the optimizer state for them.

We used standard Adam hyperparameters: $\beta_1 = 0.9, \beta_2 = 0.999, \varepsilon = 1e - 8$. For all the methods except GaLore, we selected the learning rate equal to the optimal learning rate for Adam, which we determined through a grid search among values $[1e - 4, 3e - 4, 1e - 3, 3e - 3]$. FRUGAL's learning rate for the state-free optimizer was set equal to that for the state-full optimizer for simplicity and ease of tuning. For a fair comparison with GaLore (Zhao et al., 2024a), we conducted experiments with two learning rate values: 1) the one specified by the authors in the original paper, and 2) the optimal learning rate for Adam, as used for other methods. We did this because the learning rate in the original paper could have been optimized for a different number of iterations.

To match the learning rate changes in the first steps of our training with Zhao et al. (2024a), we used a cosine learning rate schedule with restarts, with a warmup of 10% of the steps in a cycle length, and decay of the final learning rate down to 10% of the peak learning rate. To verify that our results are not sensitive to the choice of scheduler, we repeated the experiments for LLaMA-130M with other schedulers. Results for constant with warm-up and cosine (one cycle) with warm-up schedulers can be found in Tables 5 and 6.

Table 5: Perplexity of LLaMA-130M models pre-trained on C4 using constant scheduler with warm-up at various training iterations.

| Method | 100k | 200k |
|---|---|---|
| Adam | 19.51 | 18.51 |
| GaLore, $\rho = 0.25$ | 22.63 | 21.03 |
| BAdam, $\rho = 0.25$ | 22.31 | 20.66 |
| FRUGAL, $\rho = 0.25$ | **19.97** | **18.85** |
| FRUGAL, $\rho = 0.0$ | 20.33 | 19.14 |

Table 6: Perplexity of LLaMA-130M models pre-trained on C4 using cosine scheduler with warm-up at various training iterations.

| Method | 100k | 200k |
|---|---|---|
| Adam | 19.38 | 17.95 |
| GaLore, $\rho = 0.25$ | 22.30 | 20.60 |
| BAdam, $\rho = 0.25$ | 22.35 | 20.07 |
| FRUGAL, $\rho = 0.25$ | **19.62** | **18.16** |
| FRUGAL, $\rho = 0.0$ | 19.83 | 18.34 |

## A.2  FINE-TUNING SETUP

The batch size and learning rate values used for FRUGAL in the experiments from Table 4 are presented in Table 12. In all experiments, we set the learning rate for the state-free optimizer to $1/10$

Table 7: Perplexity of LLaMA-130M models pre-trained on C4 using pure bfloat16 format both for model weights and optimizer statistics.

| Method | 100k iterations |
|---|---|
| Adam | 21.88 |
| GaLore, $\rho = 0.25$ | 24.19 |
| BAdam, $\rho = 0.25$ | 25.03 |
| FRUGAL, $\rho = 0.25$ | 23.17 |
| FRUGAL, $\rho = 0.0$ | **22.64** |

Table 8: Perplexity of LLaMA-130M models pre-trained on C4 for 20k iterations (2.1B tokens) using SGD and signSGD with different learning rates. $\infty$ means that run diverged. LR stands for learning rate.

| SGD | | signSGD | |
|---|---|---|---|
| LR | Perplexity | LR | Perplexity |
| 0.1 | 184.83 | 3e-4 | 40.22 |
| 0.3 | 91.23 | 1e-3 | 41.18 |
| 1.0 | $\infty$ | 3e-3 | 109.32 |

Table 9: Perplexity of LLaMA-130M models pre-trained on C4 for 200k iterations using FRUGAL with $\rho = 1/3$ and different Block update strategy, taken from Luo et al. (2024).

| Method | Perplexity |
|---|---|
| Random | **18.50** |
| Ascending | 18.54 |
| Descending | **18.50** |

Table 10: Perplexity of LLaMA-130M models pre-trained on C4 for 200k iterations (20B tokens) using FRUGAL with $\rho = 0.25$ and different update frequency $T$.

| Update frequency $T$ | Perplexity |
|---|---|
| 10 | 18.82 |
| 20 | 18.73 |
| 50 | 18.69 |
| 100 | 18.65 |
| 200 | **18.60** |
| 500 | **18.60** |
| 1000 | 18.61 |

Table 11: Perplexity of LLaMA-130M models pre-trained on C4 for 200k iterations (20B tokens) using FRUGAL with different density $\rho$.

| | **FRUGAL** | | | | | | | |
|---|---|---|---|---|---|---|---|---|
| $\rho$ | 1.0 (**Adam**) | 0.5 | 0.33 | 0.25 | 0.125 | 0.0625 | 0.0 | **signSgd** |
| Perplexity | 18.13 | 18.40 | 18.50 | 18.63 | 18.71 | 18.80 | 18.90 | 33.22 |

of the learning rate of the state-full optimizer. Other hyperparameters, such as scheduler, number of epochs, maximum sequence length, and warmup ratio, were taken from Hu et al. (2021).

We also present a comparison between fine-tuning using FRUGAL with $\rho = 0.0$ and full fine-tuning using signSGD. Essentially, the only difference is that in the second case, the classification head is updated with signSGD instead of Adam. The results in Table 13 show that the classification head is extremely sensitive to the optimizer type, and switching the optimizer significantly drops the accuracy.

## B MEMORY ESTIMATION

In this section, we will examine memory requirements for different projection types using the LLaMA-like architecture as an example and show that RandK, column-wise, and blockwise projections result in approximately the same amount of additional memory for a given density value $\rho$ Appendix A. In contrast, the semi-orthogonal projection matrix (GaLore-like) requires a slightly larger value in this setup. Recall that we follow the setup from Zhao et al. (2024a), where Embeddings, RMSNorms, and Logits remain in the state-full subspace throughout the training, so the projection does not interact with them, and they give the same memory overhead for all projection methods.

Let the number of parameters in the remaining projectable parameters be $P$. Then, training using Adam gives an additional overhead of $2P$ float values for storing $m$ and $v$ for each parameter. Now, let's consider blockwise and column-wise projections and suppose we want to achieve a density $\rho$.

Table 12: Hyperparameters of fine-tuning RoBERTa-base for `FRUGAL`.

|  | MNLI | SST-2 | MRPC | CoLA | QNLI | QQP | RTE | STS-B |
|---|---|---|---|---|---|---|---|---|
| Batch Size | 128 | 128 | 16 | 256 | 256 | 128 | 32 | 16 |
| State-full Learning Rate | 5E-05 | 5E-05 | 2E-04 | 5E-04 | 1E-04 | 5E-05 | 2E-04 | 1E-04 |
| State-free lr multiplier | | | | 0.1 | | | | |
| Rank/Density | | | $r = 8$ $/$ $r = 0$ ($\rho = 0$) | | | | | |

Table 13: Results of fine-tuning RoBERTa-Base on several tasks from GLUE. The left column indicates which modules were trained using the state-full optimizer Adam. The remaining modules, except for the frozen Embedding layer, were trained using the state-free signSGD.

| Method | || SST2 | QNLI | QQP |
|---|---|---|---|
| **Classification head** (corresponds to the `FRUGAL` with $\rho = 0.0$) | **94.9**$_{\pm.2}$ | 92.8$_{\pm.1}$ | 91.3$_{\pm.1}$ |
| **None** (corresponds to the fine-tuning using signSGD) | 89.7 | 81.6 | 74.3 |

For blockwise, we take round($\rho \cdot L$) layers, where $L$ is the total number of transformer layers, and for column-wise, we take round($\rho \cdot k$) columns for each matrix of size $n \times k$. Since the memory required to store block or column indices is negligible compared to other costs, we find that the total size of the optimizer state when using Adam as a state-full optimizer will be $2\rho \cdot P$, with an adjustment for rounding.

In the case of RandK projection, we have the same $2\rho \cdot P$ float values $M$ and $V$ in the optimizer state. However, we must also know the current indices corresponding to these values. On the other hand, it is widely known that if one needs to save a set of random values, they don't need to store all these values - it's sufficient to store only the seed from which they were generated. Thus, for RandK, the total memory also equals $2\rho \cdot P$.

If we recalculate this considering a specific LLaMA-like architecture, each layer consists of 7 matrices: 4 matrices of size $h \times h$ (Query, Key, Value, Output) and 3 matrices of size $h \times h_{ff}$ (Gate, Down, Up), where $h$ is the hidden size of the model, and $h_{ff}$ is the FFN hidden size. In the LLaMA architecture, it's typically:

$$h_{ff} = 4h \cdot \frac{2}{3} = \frac{8}{3}h.$$

Then, the amount of memory for RandK projection (and consequently for all others mentioned above) is:

$$2 \cdot (4 \cdot (\rho h^2) + 3 \cdot (\rho \cdot h \cdot h_{ff})) = 2 \cdot (4 \cdot \rho h^2 + 3 \cdot (\frac{8}{3}\rho \cdot h^2)) = 24\rho \cdot h^2$$

for each layer on average (2 corresponds to the number of matrices $M$ and $V$).

In the case of a GaLore-like semi-orthogonal projection matrix, the situation is as follows. We have projections onto a low-rank subspace of rank $r$, where $r = $ round($\rho \cdot h$). Then, for Query, Key, Value, and Output projections, we need to store $P, M, V \in \mathbb{R}^{h \times r}$, and for Gate, Down and Up projections either $P \in \mathbb{R}^{h_{ff} \times r}, M, V \in \mathbb{R}^{h_{ff} \times r}$, or $P \in \mathbb{R}^{h_{ff} \times r}, M, V \in \mathbb{R}^{h \times r}$. Since the second option requires less memory, it is used by default in (Zhao et al., 2024a) and, therefore, in `FRUGAL`, too. Then, the total memory requirements are:

$$4 \cdot (3 \cdot rh) + 3 \cdot (2 \cdot r \cdot h + r \cdot h_{ff}) = 12rh + 6rh + 3rh_{ff} = (12 + 6 + 3 \cdot \frac{8}{3})rh = 26\rho h^2.$$

To sum up, RandK, column-wise and blockwise projection requires $2\rho P$ additional memory, while semi-orthogonal projection (GaLore-like) requires $\frac{26}{24} \cdot 2\rho P = \frac{13}{12} \cdot 2\rho P$ additional memory.

Let's recall that in addition to this, SVD requires additional computation, which can take up to 10% as the model size increases (Zhao et al., 2024a). Therefore, for our method, we settled on blockwise projection.

## C  OPTIMIZER STATE MANAGEMENT

In this section, we would like to propose some modifications to the GaLore algorithm. These modifications are also used in our framework as SVD projection.

Specifically, we want to consider the projection of the state when changing the active subspace. In GaLore (Zhao et al., 2024a), when updating the projection, the optimizer states $M$ and $V$ do not change. This results in new projected gradients and old $M$ and $V$ being in different subspaces. This implementation has little effect on the result with large values of update frequency $T$, as the values of $M$ and $V$ from the previous subspace decay exponentially quickly. However, more frequent changes $T$ significantly affect the result. We hypothesize that this is why in Zhao et al. (2024a) the model quality degraded so significantly when $T$ was decreased, while as seen in Table 10, FRUGAL experiences much less degradation.

There are two different ways to overcome this obstacle: either project the state back to full-rank space or reset the state before a new round. However, the first option may be challenging in the case of arbitrary projection. Specifically, while it's possible to project momentum back to full-rank space (see Alg. 2 in Hao et al. (2024)), the same cannot be easily done with variance because its values depend quadratically on the projection matrix. However, the projection of variance will also be trivial if the set of basis vectors for the projection is fixed, which is true, for example, for coordinate projection with RandK.

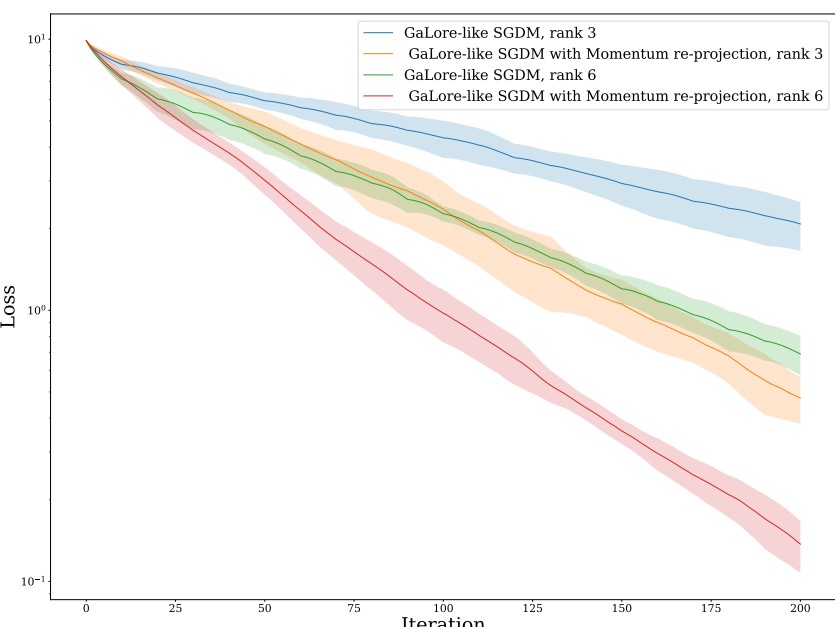

Figure 3: Toy example of solving quadratic minimization problem with GaLore-like SGDM with and without re-projection of optimizer state. Algorithm with re-projection converges much faster.

To demonstrate the effectiveness of this improvement, we provide a toy example. We consider a quadratic minimization problem of $\|W\|^2, W \in \mathbb{R}^{10 \times 10}$. For optimization, we use GaLore-like SGDM and GaLore-like SGDM with Momentum state projection. This projection is similar to Alg. 2 from (Hao et al., 2024), except we additionally normalize the new momentum by the ratio of norms before and after re-projection to preserve momentum mass. We use ranks of 3 and 6, and an update

Table 14: Pre-training LLaMA 7B on C4 dataset for 120K steps. Validation perplexity is reported.

|                          | 40K       | 80K       | 120K      |
|--------------------------|-----------|-----------|-----------|
| 8-bit Adam               | 18.09     | 15.47     | 14.83     |
| 8-bit GaLore             | 17.94     | 15.39     | 14.95     |
| FRUGAL, $\rho = 0.0$     | **17.56** | **14.50** | **13.49** |
| Tokens (B)               | 5.2       | 10.5      | 15.7      |

frequency $T = 10$ and plot mean and standard deviation across 5 independent runs. The results are presented in Figure 3. As can be seen, the variant with state projection converges much faster.

## D  LLaMA 7B PRE-TRAINING RESULTS.

In this section, we present the results of pre-training a LLaMA 7b model on the C4 dataset for 120k iterations on 12B tokens. See results in Table 14. We conducted the training in pure bfloat16 with the density $\rho = 0.0$. We used learning rate 0.0005 for state-full optimizer and 0.00015 for state-free optimizer. However, unlike Zhao et al. (2024a), we didn't use Adam8bit for state-full parameters but rather Adam, so it may not be an entirely fair comparison. Nevertheless, the results show that FRUGAL has the potential for scaling up to 7B parameter models.

# E CONVERGENCE THEORY

Firstly, we provide ommited definition of $L$-smooth function.

**Definition 1.** *We say that $f : \mathbb{R}^d \to \mathbb{R}$ is $L-$smooth with $L \geq 0$, if it is differentiable and satisfies*

$$f(y) \leq f(x) + \langle \nabla f(x), y - x \rangle + \frac{L}{2}\|y - x\|^2, \forall x, y \in \mathbb{R}^d.$$

Below, we provide an equivalent formulation of Algorithm 2 that enables us to use the proof of the similar structure to SGDM momentum analyis of Liu et al. (2020).

---

**Algorithm 3** `FRUGAL(SGDM, SGD)`: Equivalent to Algorithm 2 for constant step isze

---

**Input:** momentum weight $\beta \in [0, 1)$, initialization $x^1 \in \mathbb{R}^d$ and $m^0 = 0$, step sizes $\{\alpha_k := \alpha > 0\}_{k=1}^K$, momentum set $J_k \subset [d]$ for $k = 1, 2 \dots$.

1: **for** $k = 1, 2, \dots$ **do**
2:      Compute stochastic gradient $\tilde{g}^k \leftarrow \nabla f_{\zeta^k}(x^k)$;
3:      Update momentum vector $\tilde{m}_j^k \leftarrow (1 - \beta)\tilde{g}_j^k + \beta \begin{cases} \tilde{m}_j^{k-1} & \text{if } j \in J_k, \\ 0 & \text{otherwise}; \end{cases}$
4:      Update iterate $x^{k+1/2} \leftarrow x^k - \alpha\tilde{m}^k$;
5:      $x_j^{k+1} \leftarrow \begin{cases} \frac{x_j^{k+1/2}}{1-\beta} - \frac{\beta x_j^k}{1-\beta} & \text{if } j \notin J_{k+1}, \\ x_j^{k+1/2} & \text{otherwise}; \end{cases}$
6: **end for**

---

Next, we present several key ingredients of the proof. Firstly, we can express the momentum term $\tilde{m}_j^k$ as

$$\tilde{m}_j^k = (1 - \beta) \sum_{i=t_j^k}^{k} \beta^{k-i} \tilde{g}_j^i, \tag{4}$$

where $t_j^k := \max_{t \leq k}\{j \notin J_t\}$, i.e., the last time when the momentum buffer was released. We denote

$$m_j^k = (1 - \beta) \sum_{i=t_j^k}^{k} \beta^{k-i} g_j^i, \tag{5}$$

Using this notation, we proceed with two lemmas, one showing variance reduction effect of momentum, the other boundess of momentum bias.

**Lemma 1.** *Under Assumption 1, the update vector $\tilde{m}^k$ in Algorithm 3 satisfies*

$$\mathbb{E}\left[\|\tilde{m}^k - m^k\|^2\right] \leq \frac{1 - \beta}{1 + \beta}\sigma^2.$$

*Proof.* Since $\tilde{m}_j^k = (1 - \beta)\sum_{i=t_j^k}^k \beta^{k-i}\tilde{g}_j^i$, we have

$$\mathbb{E}\left[\|\tilde{m}^k - m^k\|^2\right] = \sum_{j\in[d]} \mathbb{E}\left[\|\tilde{m}_j^k - m_j^k\|^2\right]$$

$$\leq (1 - \beta)^2 \sum_{j\in[d]} \mathbb{E}\left[\left\|\sum_{i=t_j^k}^k \beta^{k-i}(\tilde{g}_j^i - g_j^i)\right\|^2\right].$$

Moreover, since $\zeta^1, \zeta^2, ..., \zeta^k$ are independent random variables (item 3 of Assumption 1), we can use conditional expectation to show that $\mathbb{E}\left[(\tilde{g}_j^{i_1} - g_j^{i_1})(\tilde{g}_j^{i_2} - g_j^{i_2})\right] = 0$ for $i_1 \neq i_2$. Therefore,

$$\mathbb{E}\left[\left\|\tilde{m}^k - m^k\right\|^2\right] \le (1-\beta)^2 \sum_{j\in[d]} \mathbb{E}\left[\sum_{i=t_j^k}^k \beta^{2(k-i)}\|\tilde{g}_j^i - g_j^i\|^2\right]$$

$$\le \frac{1-\beta}{1+\beta}\sum_{j\in[d]}\mathbb{E}\left[(1-\beta^{2(k-t_j^k+1)})\right]\sigma_j^2$$

$$\le \frac{1-\beta}{1+\beta}\sum_{j\in[d]}\sigma_j^2 = \frac{1-\beta}{1+\beta}\sigma^2.$$

$\square$

**Lemma 2.** *Under Assumption 1, the update vector $\tilde{m}^k$ in Algorithm 3 further satisfies*

$$\mathbb{E}\left[\sum_{j\in J_k}(1-\beta^{k_j})^2\left\|\frac{m_j^k}{(1-\beta^{k_j})} - g_j^k\right\|^2\right] \le p_{\max}^k\mathbb{E}\left[\sum_{i=1}^{k-1}a_{k,i}\|x^{i+1}-x^i\|^2\right],$$

*where $k_j = k - t_j^k + 1$, and*

$$a_{k,i} = L^2\beta^{k-i}\left(k-i+\frac{\beta}{1-\beta}\right). \tag{6}$$

*Proof.* Let $\Pr_{k-1}[j\in J_k] = p_j^k$ and $p_{\max}^k := \max_{j\in[d]}\{p_j^k\}$. Then,

$$\mathbb{E}\left[\sum_{j\in J_k}(1-\beta^{k_j})^2\left\|\frac{m_j^k}{(1-\beta^{k_j})} - g_j^k\right\|^2\right] = \mathbb{E}\left[\sum_{j\in J_k}(1-\beta^{k_j})^2\left\|\frac{1-\beta}{1-\beta^{k_j}}\sum_{i=t_j^k}^k\beta^{k-i}(g_j^i - g_j^k)\right\|^2\right]$$

$$= (1-\beta)^2\mathbb{E}\left[\sum_{j\in J_k}\sum_{i,l=t_j^k}^k\langle\beta^{k-i}(g_j^k-g_j^i),\beta^{k-l}(g_j^k-g_j^l)\rangle\right]$$

$$\le (1-\beta)^2\mathbb{E}\left[\sum_{j\in J_k}\sum_{i,l=1}^k\beta^{2k-i-l}\left(\frac{1}{2}\|g_j^k-g_j^i\|^2] + \frac{1}{2}\|g_j^k-g_j^l\|^2\right)\right]$$

$$= (1-\beta)^2\mathbb{E}\left[\sum_{j\in J_k}\sum_{i=1}^k\left(\sum_{l=1}^k\beta^{2k-i-l}\right)\frac{1}{2}\mathbb{E}[\|g_j^k-g_j^l\|^2]\right]$$

$$+ (1-\beta)^2\mathbb{E}\left[\sum_{j\in J_k}\sum_{l=1}^k\left(\sum_{i=1}^k\beta^{2k-i-l}\right)\frac{1}{2}[\|g_j^k-g_j^i\|^2]\right]$$

$$= (1-\beta)^2\mathbb{E}\left[\sum_{j\in J_k}\sum_{i=1}^k\frac{\beta^{k-i}(1-\beta^{k_j})}{1-\beta}\|g_j^k-g_j^i\|^2\right]$$

$$\le (1-\beta)\mathbb{E}\left[\sum_{j\in J_k}\sum_{i=1}^k\beta^{k-i}\|g_j^k-g_j^i\|^2\right],$$

$$\le (1-\beta)p_{\max}^k\mathbb{E}\left[\sum_{i=1}^k\beta^{k-i}\|g^k-g^i\|^2\right],$$

where we applied Cauchy-Schwarz to the first inequality.

By applying triangle inequality and the smoothness of $f$ (item 1 in Assumption 1), we further have

$$\mathbb{E}\left[\sum_{j \in J_k}(1-\beta^{k_j})^2 \left\|\frac{m_j^k}{(1-\beta^{k_j})} - g_j^k\right\|^2\right] \leq (1-\beta)p_{\max}^k \mathbb{E}\left[\sum_{i=1}^k \beta^{k-i}(k-i)\sum_{l=i}^{k-1}\|g^{l+1}-g^l\|^2\right]$$

$$\leq \mathbb{E}\left[\sum_{l=1}^{k-1}\left((1-\beta)p_{\max}^k L^2 \sum_{i=1}^l \beta^{k-i}(k-i)\right)\|x^{l+1}-x^l\|^2\right].$$

Therefore, by defining $a'_{k,l} = (1-\beta)L^2 \sum_{i=1}^l \beta^{k-i}(k-i)$, we get

$$\mathbb{E}\left[\sum_{j \in J_k}(1-\beta^{k_j})^2 \left\|\frac{m_j^k}{(1-\beta^{k_j})} - g_j^k\right\|^2\right] \leq p_{\max}^k \mathbb{E}\left[\sum_{l=1}^{k-1} a'_{k,l}\|x^{l+1}-x^l\|^2\right]. \tag{7}$$

Furthermore, $a'_{k,j}$ can be calculated as

$$a'_{k,l} = L^2\beta^k\left(-(k-1)-\frac{1}{1-\beta}\right) + L^2\beta^{k-l}\left(k-l+\frac{\beta}{1-\beta}\right). \tag{8}$$

Notice that

$$a'_{k,l} < a_{k,l} := L^2\beta^{k-l}\left(k-l+\frac{\beta}{1-\beta}\right). \tag{9}$$

Combining this with equation 7, we arrive at

$$\mathbb{E}\left[\sum_{j \in J_k}(1-\beta^{k_j})^2 \left\|\frac{m_j^k}{(1-\beta^{k_j})} - g_j^k\right\|^2\right] \leq p_{\max}^k \mathbb{E}\left[\sum_{i=1}^{k-1} a_{k,i}\|x^{i+1}-x^i\|^2\right],$$

where

$$a_{k,i} = L^2\beta^{k-i}\left(k-i+\frac{\beta}{1-\beta}\right).$$

$\square$

From Lemma 2, we know that the distance of the non-stochastic momentum from $g^k$ is bounded by the weighted sum of past successive iterate differences. Furthermore, the coefficients $a_{k,i}$ decays exponentially in $\beta$.

Therefore, we use the following Lyapunov function

$$L^k = \left(f(z^k) - f^\star\right) + \sum_{i=1}^{k-1} c_i\|x^{k+1-i} - x^{k-i}\|^2. \tag{10}$$

for some positive $c_i$ that we specify later. As it is common for convergence theory of SGDM to analyze an auxiliary sequence $z^k$ defined as

$$z_j^k = \begin{cases} x_j^k & k=1, \\ \frac{1}{1-\beta}x_j^{k-1/2} - \frac{\beta}{1-\beta}x_j^{k-1} & k \geq 2, \end{cases} \tag{11}$$

which behaves more like an SGD iterate, although the stochastic gradient $\tilde{g}^k$ is not taken at $z^k$.

**Lemma 3.** *Let $x^k$'s be iterates of Algorithm 3, then $z^k$ defined in equation 11 satisfies*

$$z^{k+1} - z^k = -\alpha\tilde{g}^k.$$

*Proof.* We have to consider two different cases. Firstly, if $k=1$ or $j \notin J_k$, then

$$z_j^{k+1} - z_j^k = \frac{x_j^{k+1/2}}{1-\beta} - \frac{\beta x_j^k}{1-\beta} - x_j^k = \frac{x_j^k - \alpha\tilde{m}_j^k - \beta x_j^k - (1-\beta)x_j^k}{1-\beta} = -\frac{\alpha(1-\beta)\tilde{g}_j^k}{1-\beta} = -\alpha\tilde{g}_j^k.$$

Secondly, if $k \geq 2$, $j \in J_k$, then

$$
\begin{aligned}
z_j^{k+1} - z_j^k &= \frac{1}{1-\beta}(x_j^{k+1/2} - x_j^{k-1/2}) - \frac{\beta}{1-\beta}(x_j^k - x_j^{k-1}) \\
&= \frac{1}{1-\beta}(x_j^{k+1/2} - x_j^k) - \frac{\beta}{1-\beta}(x_j^k - x_j^{k-1}) \\
&= \frac{1}{1-\beta}(-\alpha \tilde{m}_j^k) - \frac{\beta}{1-\beta}(-\alpha \tilde{m}_j^{k-1}) \\
&= \frac{1}{1-\beta}(-\alpha \tilde{m}_j^k + \alpha \beta \tilde{m}_j^{k-1}) = -\alpha \tilde{g}_j^k.
\end{aligned}
$$

$\square$

Before procceding with the main convergence theory, we require one more proposition that shows descent in objective value.

**Proposition 1.** *Take Assumption 1. Then, for $z^k$ defined in equation 11, we have*

$$
\mathbb{E}[f(z^{k+1})] \leq \mathbb{E}[f(z^k)] + \left(-\alpha + \frac{1+\beta^2}{1-\beta}L\alpha^2 + \frac{1}{2}L\alpha^2\right)\mathbb{E}[\|g^k\|^2]
$$

$$
+ \left(\frac{\beta^2}{2(1+\beta)} + \frac{1}{2}\right)L\alpha^2\sigma^2 + \frac{L\alpha^2}{1-\beta}\mathbb{E}\left[\sum_{j \in J_k}(1-\beta^{k_j})^2\left\|\frac{m_j^k}{(1-\beta^{k_j})} - g_j^k\right\|^2\right]. \tag{12}
$$

*Proof.* The smoothness of $f$ yields

$$
\begin{aligned}
\mathbb{E}_{\zeta^k}[f(z^{k+1})] &\leq f(z^k) + \mathbb{E}_{\zeta^k}[\langle \nabla f(z^k), z^{k+1} - z^k\rangle] + \frac{L}{2}\mathbb{E}_{\zeta^k}[\|z^{k+1} - z^k\|^2] \\
&= f(z^k) + \mathbb{E}_{\zeta^k}[\langle \nabla f(z^k), -\alpha\tilde{g}^k\rangle] + \frac{L\alpha^2}{2}\mathbb{E}_{\zeta^k}[\|\tilde{g}^k\|^2],
\end{aligned} \tag{13}
$$

where we have applied Lemma 3 in the second step.

For the inner product term, we can take full expectation $\mathbb{E} = \mathbb{E}_{\zeta^1}...\mathbb{E}_{\zeta^k}$ to get

$$
\mathbb{E}[\langle \nabla f(z^k), -\alpha\tilde{g}^k\rangle] = \mathbb{E}[\langle \nabla f(z^k), -\alpha g^k\rangle],
$$

which follows from the fact that $z^k$ is determined by the previous $k-1$ random samples $\zeta^1, \zeta^2, ...\zeta^{k-1}$, which is independent of $\zeta^k$, and $\mathbb{E}_{\zeta^k}[\tilde{g}^k] = g^k$.

So, we can bound

$$
\begin{aligned}
\mathbb{E}[\langle \nabla f(z^k), -\alpha\tilde{g}^k\rangle] &= \mathbb{E}[\langle \nabla f(z^k) - g^k, -\alpha g^k\rangle] - \alpha\mathbb{E}[\|g^k\|^2] \\
&\leq \alpha\frac{\rho_0}{2}L^2\mathbb{E}[\|z^k - x^k\|^2] + \alpha\frac{1}{2\rho_0}\mathbb{E}[\|g^k\|^2] - \alpha\mathbb{E}[\|g^k\|^2],
\end{aligned}
$$

where $\rho_0 > 0$ can be any positive constant (to be determined later).

Combining equation 13 and the last inequality, we arrive at

$$
\begin{aligned}
\mathbb{E}[f(z^{k+1})] &\leq \mathbb{E}[f(z^k)] + \alpha\frac{\rho_0}{2}L^2\mathbb{E}[\|z^k - x^k\|^2] \\
&\quad + (\alpha\frac{1}{2\rho_0} - \alpha)\mathbb{E}[\|g^k\|^2] + \frac{L\alpha^2}{2}\mathbb{E}[\|\tilde{g}^k\|^2].
\end{aligned}
$$

By construction, $z_j^k - x_j^k = -\frac{\beta}{1-\beta}\alpha\tilde{m}_j^{k-1}$ for $j \in J_k$, 0 otherwise. Consequently,

$$
\begin{aligned}
\mathbb{E}[f(z^{k+1})] &\leq \mathbb{E}[f(z^k)] + \alpha^3\frac{\rho_0}{2}L^2(\frac{\beta}{1-\beta})^2\mathbb{E}\left[\sum_{j \in J_k}\|\tilde{m}_j^{k-1}\|^2\right] \\
&\quad + (\alpha\frac{1}{2\rho_0} - \alpha)\mathbb{E}[\|g^k\|^2] + \frac{L\alpha^2}{2}\mathbb{E}[\|\tilde{g}^k\|^2].
\end{aligned} \tag{14}
$$

Let $k_j = k - t_j^{k-1} + 1$. Then, from Lemma 1 we know that

$$\mathbb{E}\left[\sum_{j \in J_k} \|\tilde{m}_j^{k-1}\|^2\right] \leq 2\mathbb{E}\left[\sum_{j \in J_k} \|\tilde{m}_j^{k-1} - m_j^{k-1}\|^2\right] + 2\mathbb{E}\left[\sum_{j \in J_k} \|m_j^{k-1}\|^2\right]$$

$$\leq 2\frac{1-\beta}{1+\beta}\mathbb{E}\left[\sum_{j \in J_k} \sigma_j^2 + 2\sum_{j \in J_k} \|m_j^{k-1}\|^2\right]$$

$$\mathbb{E}\left[\sum_{j \in J_k} \|m_j^{k-1}\|^2\right] = \mathbb{E}\left[\sum_{j \in J_k} (1 - \beta^{(k-1)_j})^2 \left\|\frac{m_j^{k-1}}{(1 - \beta^{(k-1)_j})}\right\|^2\right]$$

$$\leq 2\mathbb{E}\left[\sum_{j \in J_k} (1 - \beta^{(k-1)_j})^2 \left\|\frac{m_j^{k-1}}{(1 - \beta^{(k-1)_j})} - g_j^k\right\|^2\right] + 2\mathbb{E}\left[\sum_{j \in J_k} \|g_j^k\|^2\right]$$

$$\mathbb{E}\left[\|\tilde{g}^k\|^2\right] \leq \sigma^2 + \mathbb{E}[\|g^k\|^2]. \tag{15}$$

Putting these into equation 14, we arrive at

$$\mathbb{E}[f(z^{k+1})] \leq \mathbb{E}[f(z^k)] + \left(-\alpha + \alpha\frac{1}{2\rho_0} + 2\alpha^3\rho_0 L^2\left(\frac{\beta}{1-\beta}\right)^2 + \frac{L\alpha^2}{2}\right)\mathbb{E}[\|g^k\|^2]$$

$$+ \left(\alpha^3\rho_0 L^2\left(\frac{\beta}{1-\beta}\right)^2\frac{1-\beta}{1+\beta}\sigma^2 + \frac{L\alpha^2}{2}\sigma^2\right)$$

$$+ 2\alpha^3\rho_0 L^2\left(\frac{\beta}{1-\beta}\right)^2 \mathbb{E}\left[\sum_{j \in J_k} (1 - \beta^{(k-1)_j})^2 \left\|\frac{m_j^{k-1}}{(1 - \beta^{(k-1)_j})} - g_j^k\right\|^2\right].$$

Notice that if $j \in J^k$, then $(k-1)_j = k_j - 1$. Therefore,

$$\mathbb{E}\left[\left\|\frac{m_j^k}{(1-\beta^{k_j})} - g_j^k\right\|^2\right] = \mathbb{E}\left[\left\|\frac{\beta m_j^{k-1} + (1-\beta)g_j^k}{(1-\beta^{k_j})} - g_j^k\right\|^2\right]$$

$$= \beta^2\mathbb{E}\left[\left(\frac{(1-\beta^{k_j-1})}{(1-\beta^{k_j})}\right)^2 \left\|\frac{m_j^{k-1}}{(1-\beta^{(k-1)_j})} - g_j^k\right\|^2\right].$$

Substituting the above into the last inequality produces

$$\mathbb{E}[f(z^{k+1})] \leq \mathbb{E}[f(z^k)] + \left(-\alpha + \alpha\frac{1}{2\rho_0} + 2\alpha^3\rho_0 L^2(\frac{\beta}{1-\beta})^2 + \frac{L\alpha^2}{2}\right)\mathbb{E}[\|g^k\|^2]$$

$$+ \left(\alpha^3\rho_0 L^2(\frac{\beta}{1-\beta})^2\frac{1-\beta}{1+\beta}\sigma^2 + \frac{L\alpha^2}{2}\sigma^2\right) \tag{16}$$

$$+ 2\alpha^3\rho_0 L^2\left(\frac{1}{1-\beta}\right)^2 \mathbb{E}\left[\sum_{j \in J_k} (1 - \beta^{k_j})^2 \left\|\frac{m_j^k}{(1 - \beta^{k_j})} - g_j^k\right\|^2\right].$$

Finally, $\rho_0 = \frac{1-\beta}{2L\alpha}$ gives

$$\mathbb{E}[f(z^{k+1})] \leq \mathbb{E}[f(z^k)] + \left(-\alpha + \frac{1+\beta^2}{1-\beta}L\alpha^2 + \frac{1}{2}L\alpha^2\right)\mathbb{E}[\|g^k\|^2]$$

$$+ \left(\frac{\beta^2}{2(1+\beta)} + \frac{1}{2}\right)L\alpha^2\sigma^2 + \frac{L\alpha^2}{1-\beta}\mathbb{E}\left[\sum_{j \in J_k} (1 - \beta^{k_j})^2 \left\|\frac{m_j^k}{(1 - \beta^{k_j})} - g_j^k\right\|^2\right].$$

$\square$

### E.1 CONVERGENCE OF ALGORITHM 3

Firstly, by combining results from prior section, we can bound our Lyapunov function $L^k$ defined in equation 10.

**Proposition 2.** *Let Assumption 1 hold and $\alpha \leq \frac{1-\beta}{2\sqrt{2}L\sqrt{p_{\max}^k}\sqrt{\beta+\beta^2}}$ in Algorithm 3. Let $\{c_i\}_{i=1}^{\infty}$ in equation 10 be defined by*

$$c_1 = \frac{\frac{\beta+\beta^2}{(1-\beta)^3}L^3\alpha^2}{1-4\alpha^2\frac{\beta+\beta^2}{(1-\beta)^2}L^2}, \qquad c_{i+1} = c_i - \left(4c_1\alpha^2 + \frac{L\alpha^2}{1-\beta}\right)\beta^i(i+\frac{\beta}{1-\beta})L^2 \quad \text{for all } i \geq 1.$$

*Then, $c_i > 0$ for all $i \geq 1$, and*

$$
\mathbb{E}[L^{k+1} - L^k] \leq \left(-\alpha + \frac{3-\beta+\beta^2}{2(1-\beta)}L\alpha^2 + 4c_1\alpha^2\right)\mathbb{E}[\|g^k\|^2] \tag{17}
$$
$$
+ \left(\frac{\beta^2}{2(1+\beta)}L\alpha^2\sigma^2 + \frac{1}{2}L\alpha^2\sigma^2 + 2c_1\alpha^2\sigma^2\right).
$$

*Proof.* Recall that $L^k$ is defined as

$$L^k = f(z^k) - f^* + \sum_{i=1}^{k-1} c_i\|x^{k+1-i} - x^{k-i}\|^2,$$

Therefore, by equation 16 we know that

$$\mathbb{E}[L^{k+1} - L^k] \leq$$
$$(-\alpha + \frac{1+\beta^2}{1-\beta}L\alpha^2 + \frac{1}{2}L\alpha^2)\mathbb{E}[\|g^k\|^2]$$
$$+ \sum_{i=1}^{k-1}(c_{i+1} - c_i)\mathbb{E}[\|x^{k+1-i} - x^{k-i}\|^2] + c_1\mathbb{E}[\|x^{k+1} - x^k\|^2] \tag{18}$$
$$+ \left(\frac{\beta^2}{2(1+\beta)} + \frac{1}{2}\right)L\alpha^2\sigma^2 + \frac{L\alpha^2}{1-\beta}\mathbb{E}\left[\sum_{j\in J_k}(1-\beta^{k_j})^2\left\|\frac{m_j^k}{(1-\beta^{k_j})} - g_j^k\right\|^2\right].$$

To bound the $c_1\mathbb{E}[\|x^{k+1} - x^k\|^2]$ term, we need the following inequalities, which are obtained similarly as equation 15.

$$\mathbb{E}[\|\tilde{m}^k\|^2] \leq 2\frac{1-\beta}{1+\beta}\sigma^2 + 2\mathbb{E}[\|m^k\|^2]$$
$$\mathbb{E}[\|m^k\|^2] \leq 2\mathbb{E}\left[\sum_{j\in J_k}(1-\beta^{k_j})^2\left\|\frac{m_j^k}{(1-\beta^{k_j})} - g_j^k\right\|^2\right] + 2\mathbb{E}\left[\|g^k\|^2\right] \tag{19}$$
$$\mathbb{E}[\|\tilde{g}^k\|^2] \leq \sigma^2 + \mathbb{E}[\|g^k\|^2].$$

Let $\Pr_{k-1}[j \in J_k] = p_j^k$ and $p_{\min}^k := \min_{j\in[d]}\{p_j^k\}$. Then, $c_1\mathbb{E}[\|x^{k+1} - x^k\|^2]$ can be bounded as

$$c_1\mathbb{E}[\|x^{k+1} - x^k\|^2] = c_1\alpha^2\mathbb{E}[\|\tilde{u}^k\|^2] = c_1\alpha^2\mathbb{E}\left[\sum_{j\in J_k}\|\tilde{m}_j^k\|^2 + \sum_{j\notin J_k}\|\tilde{g}_j^k\|^2\right]$$
$$\leq c_1\alpha^2\mathbb{E}\left[\|\tilde{m}^k\|^2 + (1-p_{\min}^k)\|\tilde{g}^k\|^2\right]$$
$$\leq c_1\alpha^2\left(\left(2\frac{1-\beta}{1+\beta} + 1 - p_{\min}^k\right)\sigma^2 + 5\mathbb{E}[\|g^k\|^2]\right)$$
$$+ 4c_1\alpha^2\mathbb{E}\left[\sum_{j\in J_k}(1-\beta^{k_j})^2\left\|\frac{m_j^k}{(1-\beta^{k_j})} - g_j^k\right\|^2\right]$$

Combine this with equation 18, we obtain

$$\mathbb{E}[L^{k+1} - L^k]$$

$$\leq (-\alpha + \frac{1+\beta^2}{1-\beta}L\alpha^2 + \frac{1}{2}L\alpha^2 + 5c_1\alpha^2)\mathbb{E}[\|g^k\|^2] + \left(\frac{\beta^2}{2(1+\beta)} + \frac{1}{2} + \frac{c_1}{L}\left(2\frac{1-\beta}{1+\beta} + 1 - p_{\min}^k\right)\right)L\alpha^2\sigma^2$$

$$+ \sum_{i=1}^{k-1}(c_{i+1} - c_i)\mathbb{E}[\|x^{k+1-i} - x^{k-i}\|^2]$$

$$+ \left(4c_1\alpha^2 + \frac{L\alpha^2}{1-\beta}\right)\mathbb{E}\left[\sum_{j\in J_k}(1 - \beta^{k_j})^2\left\|\frac{m_j^k}{(1-\beta^{k_j})} - g_j^k\right\|^2\right].$$

$$(20)$$

In the rest of the proof, let us show that the sum of the last two terms in equation 20 is non-positive.

First of all, by Lemma 2 we know that

$$\mathbb{E}\left[\sum_{j\in J_k}(1 - \beta^{k_j})^2\left\|\frac{m_j^k}{(1-\beta^{k_j})} - g_j^k\right\|^2\right] \leq \mathbb{E}\left[p_{\max}^k\sum_{i=1}^{k-1}a_{k,i}\|x^{i+1} - x^i\|^2\right],$$

where

$$a_{k,i} = L^2\beta^{k-i}\left(k - i + \frac{\beta}{1-\beta}\right).$$

Or equivalently,

$$\mathbb{E}\left[\sum_{j\in J_k}(1 - \beta^{k_j})^2\left\|\frac{m_j^k}{(1-\beta^{k_j})} - g_j^k\right\|^2\right] \leq \mathbb{E}\left[\sum_{i=1}^{k-1}p_{\max}^k a_{k,k-i}\|x^{k+1-i} - x^{k-i}\|^2\right],$$

where

$$a_{k,k-i} = L^2\beta^i\left(i + \frac{\beta}{1-\beta}\right).$$

Therefore, to make the sum of the last two terms of equation 20 to be non-positive, we need to have

$$c_{i+1} \leq c_i - \left(4c_1\alpha^2 + \frac{L\alpha^2}{1-\beta}\right)L^2 p_{\max}^i\beta^i\left(i + \frac{\beta}{1-\beta}\right)$$

for all $i \geq 1$. To satisfy this inequality, we choose

$$c_{i+1} = c_i - \left(4c_1\alpha^2 + \frac{L\alpha^2}{1-\beta}\right)L^2\beta^i p_{\max}^i\left(i + \frac{\beta}{1-\beta}\right)$$

for all $i \geq 1$, which implies that

$$c_i = c_1 - \left(4c_1\alpha^2 + \frac{L\alpha^2}{1-\beta}\right)L^2\sum_{l=1}^{i-1}\beta^i p_{\max}^i\left(i + \frac{\beta}{1-\beta}\right).$$

To have $c_i > 0$ for all $i \geq 1$, we can set $c_1$ as

$$c_1 = \left(4c_1\alpha^2 + \frac{L\alpha^2}{1-\beta}\right)L^2\hat{p}_{\max}^k\sum_{i=1}^{\infty}\beta^i\left(i + \frac{\beta}{1-\beta}\right).$$

where, $\hat{p}_{\max}^k = \max_{i\in[k]}\{p_{\max}^i\}$. Since

$$\sum_{i=1}^{j}i\beta^i = \frac{1}{1-\beta}\left(\frac{\beta(1-\beta^j)}{1-\beta} - j\beta^{j+1}\right),$$

we have $\sum_{i=1}^{\infty} i\beta^i = \frac{\beta}{(1-\beta)^2}$ and

$$c_1 = \left(4c_1\alpha^2 + \frac{L\alpha^2}{1-\beta}\right)L^2\hat{p}_{\max}^k\frac{\beta+\beta^2}{(1-\beta)^2},$$

which implies that

$$c_1 = \frac{\alpha^2 L^3\hat{p}_{\max}^k\frac{\beta+\beta^2}{(1-\beta)^3}}{1 - 4\alpha^2\frac{\beta+\beta^2}{(1-\beta)^2}\hat{p}_{\max}^k L^2}. \tag{21}$$

Notice that $\alpha \leq \frac{1-\beta}{2\sqrt{2}L\sqrt{\hat{p}_{\max}^k}\sqrt{\beta+\beta^2}}$ ensures $c_1 > 0$.

Therefore,

$$\mathbb{E}[L^{k+1} - L^k] \leq \left(-\alpha + \frac{3-\beta+2\beta^2}{2(1-\beta)}L\alpha^2 + 5c_1\alpha^2\right)\mathbb{E}[\|g^k\|^2]$$

$$+ \left(\frac{\beta^2}{2(1+\beta)}L\alpha^2\sigma^2 + \frac{1}{2}L\alpha^2\sigma^2 + c_1\alpha^2\sigma^2\left(2\frac{1-\beta}{1+\beta} + 1 - p_{\min}^k\right)\right).$$

$\square$

By telescoping equation 17, we obtain the convergence bound of our proposed algorithm under nonconvex settings.

**Theorem 2.** *Let Assumption 1 hold and $\alpha^k = \alpha \leq \frac{1-\beta}{L(4-\beta+\beta^2)}$. Then, the iterates of Algorithm 3 satisfy*

$$\frac{1}{k}\sum_{i=1}^{k}\mathbb{E}[\|g^i\|^2] \leq \mathcal{O}\left(\frac{f(x^1) - f^*}{k\alpha} + L\alpha\sigma^2\left(1 + \frac{\hat{p}_{\max}^k(1-\bar{p}_{\min}^k)\beta}{(1-\beta)}\right)\right), \tag{22}$$

*where $\bar{p}_{\min}^k = \frac{1}{k}\sum_{i=1}^{k}\bar{p}_{\min}^i$ and $\hat{p}_{\max}^k = \max_{i\in[k]}\{p_{\max}^i\}$.*

*Proof.* From equation 17 we know that

$$\mathbb{E}[L^{k+1} - L^k] \leq -R_1\mathbb{E}[\|g^k\|^2] + R_2^k, \tag{23}$$

where

$$R_1 = -\alpha + \frac{3-\beta+\beta^2}{2(1-\beta)}L\alpha^2 + 4c_1\alpha^2,$$

$$R_2 = \frac{\beta^2}{2(1+\beta)}L\alpha^2\sigma^2 + \frac{1}{2}L\alpha^2\sigma^2 + c_1\alpha^2\sigma^2\left(2\frac{1-\beta}{1+\beta} + 1 - p_{\min}^k\right).$$

We further define

$$\bar{R}_2 = \frac{\beta^2}{2(1+\beta)}L\alpha^2\sigma^2 + \frac{1}{2}L\alpha^2\sigma^2 + c_1\alpha^2\sigma^2\left(2\frac{1-\beta}{1+\beta} + 1 - \bar{p}_{\min}^k\right),$$

where $\bar{p}_{\min}^k = \frac{1}{k}\sum_{i=1}^{k}\bar{p}_{\min}^i$.

Telescoping equation 23 yields

$$L^1 \geq \mathbb{E}[L^1 - L^{k+1}] \geq R_1\sum_{i=1}^{k}\mathbb{E}[\|g^i\|^2] - \sum_{k=1}^{k}R_2^k,$$

and therefore

$$\frac{1}{k}\sum_{i=1}^{k}\mathbb{E}[\|g^i\|^2] \leq \frac{L^1}{kR_1} + \frac{\bar{R}_2}{R_1}. \tag{24}$$

In the rest of the proof, we will appropriately bound $R_1$ and $\bar{R}_2$.

First, let us show that $R_1 \geq \frac{\alpha}{2}$ and $\alpha \leq \min\left\{ \frac{1-\beta}{L(4-\beta+\beta^2)}, \frac{1-\beta}{2\sqrt{2}L\sqrt{\hat{p}_{\max}^k}\sqrt{\beta+\beta^2}} \right\}$.

From equation 21 we know that

$$c_1 = \frac{\alpha^2 L^3 \hat{p}_{\max}^k \frac{\beta+\beta^2}{(1-\beta)^3}}{1 - 4\alpha^2 \frac{\beta+\beta^2}{(1-\beta)^2} L^2 \hat{p}_{\max}^k}.$$

Since $\alpha \leq \frac{1-\beta}{2\sqrt{2}L\sqrt{\hat{p}_{\max}^k}\sqrt{\beta+\beta^2}}$, we have

$$4\alpha^2 \frac{\beta+\beta^2}{(1-\beta)^2} L^2 \hat{p}_{\max}^k \leq \frac{1}{2}.$$

Thus,

$$c_1 \leq \alpha^2 L^3 \hat{p}_{\max}^k \frac{\beta+\beta^2}{(1-\beta)^3} \leq \frac{L}{8(1-\beta)}.$$

Therefore, in order to ensure $R_1 \geq \frac{\alpha}{2}$, it suffices to have

$$\frac{3-\beta+\beta^2}{2(1-\beta)} L\alpha + \frac{\alpha L}{2(1-\beta)} \leq \frac{1}{2}$$

which is equivalent to our condition $\alpha \leq \frac{1-\beta}{L(4-\beta+\beta^2)}$.

For $\bar{R}_2$, we can upperbound $c_1$ using our condition $\alpha \leq \frac{1-\beta}{L(4-\beta+\beta^2)}$. Thus,

$$c_1 \leq \alpha^2 L^3 \hat{p}_{\max}^k \frac{\beta+\beta^2}{(1-\beta)^3} \leq \frac{\hat{p}_{\max}^k \beta L}{2(1-\beta)}.$$

Therefore,

$$\bar{R}_2 = \frac{\beta^2}{2(1+\beta)} L\alpha^2\sigma^2 + \frac{1}{2}L\alpha^2\sigma^2 + c_1\alpha^2\sigma^2 \left( 2\frac{1-\beta}{1+\beta} + 1 - \bar{p}_{\min}^k \right)$$

$$\leq \frac{\beta^2}{2(1+\beta)} L\alpha^2\sigma^2 + \frac{1}{2}L\alpha^2\sigma^2 + \frac{\hat{p}_{\max}^k \beta L\alpha^2\sigma^2}{(1+\beta)} + L\alpha^2\sigma^2 \hat{p}_{\max}^k (1-\bar{p}_{\min}^k)\frac{\beta}{1-\beta}$$

$$\leq \left( \frac{2\beta^2 + 8\hat{p}_{\max}^k}{2(1+\beta)} + \frac{1}{2} + \frac{\hat{p}_{\max}^k(1-\bar{p}_{\min}^k)\beta}{8(1-\beta)} \right) L\alpha^2\sigma^2.$$

By putting them all together, we obtain

$$\frac{1}{k}\sum_{i=1}^{k} \mathbb{E}[\|g^i\|^2] \leq \frac{2\left(f(x^1) - f^*\right)}{k\alpha} + \left( \frac{2\beta^2 + 8\hat{p}_{\max}^k}{2(1+\beta)} + \frac{1}{2} + \frac{\hat{p}_{\max}^k(1-\bar{p}_{\min}^k)\beta}{8(1-\beta)} \right) L\alpha\sigma^2$$

$$= \mathcal{O}\left( \frac{f(x^1) - f^*}{k\alpha} + L\alpha\sigma^2 \left( 1 + \frac{\hat{p}_{\max}^k(1-\bar{p}_{\min}^k)\beta}{(1-\beta)} \right) \right).$$

$\square$

Table 15: Comparison of different projection and state-free subspace optimization strategies for different values density $\rho$ on pre-training LLaMA-130M on C4 with Adam as the state-full algorithm.

| Density $\rho$ | Projection type | Optimizes state-free subspace | Validation perplexity after iterations ↓ | | | | |
|---|---|---|---|---|---|---|---|
| | | | 4k | 20k | 40k | 100k | 200k |
| 0.5 | SVD | No | **36.15** | **23.85** | 22.09 | 20.32 | 19.30 |
| | Random | No | 38.52 | 23.91 | **21.89** | **19.97** | **18.90** |
| | Blockwise | Yes | 34.80 | 22.59 | 21.27 | 19.38 | 18.40 |
| | SVD | Yes | **34.18** | **22.45** | **20.85** | **19.23** | **18.30** |
| 0.333 | SVD | No | **38.00** | 25.18 | 23.31 | 21.42 | 20.31 |
| | Random | No | 40.30 | **25.00** | **22.78** | **20.65** | **19.46** |
| | Blockwise | Yes | 35.77 | 22.81 | 21.28 | 19.50 | 18.50 |
| | SVD | Yes | **34.33** | **22.54** | **20.91** | **19.25** | **18.33** |
| 0.125 | SVD | No | **44.48** | 29.24 | 26.80 | 24.37 | 22.91 |
| | Random | No | 48.65 | **28.90** | **25.78** | **22.94** | **21.35** |
| | Blockwise | Yes | 37.21 | 23.70 | 21.69 | 19.76 | 18.71 |
| | SVD | Yes | **34.95** | **22.83** | **21.16** | **19.44** | **18.48** |
| 0.0625 | SVD | No | **51.05** | **33.01** | 29.88 | 26.84 | 25.07 |
| | Random | No | 60.54 | 35.64 | **29.02** | **25.30** | **23.41** |
| | Blockwise | Yes | 37.94 | 23.54 | 21.53 | 19.90 | 18.80 |
| | SVD | Yes | **35.18** | **22.93** | **21.24** | **19.56** | **18.56** |

# F ADDITIONAL EXPERIMENTS

In this section we present additional experiments.

**Connection between density and type of the projection.** First, we present the results of the experiments that follow the setup of Table 1 but explore different density $\rho$ values: 0.0625, 0.125, 0.333, and 0.5 (while in Table 1 we use $\rho = 0.25$). The results, presented in Table 15, align with our findings from Table 1. Specifically, training with random projection significantly outperforms SVD projection when training without optimizing the state-free subspace. When state-free subspace optimization is employed, SVD projections marginally outperform their Blockwise counterparts.

**Different state-full and state-free optimizers.** Next, we conducted experiments for other state-full and state-free optimizers. We explored two variations: 1. replacing AdamW with Lion (Chen et al., 2024) as the state-full optimizer, and 2. substituting signSGD with SGD as the state-free optimizer. We pre-trained LLaMA-130M on C4 for 200k steps with the hyperparameters specified in Appendix A.1. We approached the Lion experiments in the same way as the Adam experiments: first finding the optimal learning rate for the original algorithm through grid search, then using that same learning rate for both GaLore and FRUGAL. For SGD experiments, we kept state-full Adam's learning rate constant while only adjusting the learning rate for the state-free optimizer.

The results are presented in tables Table 16 and Table 17. As observed, the results for Lion are similar to those obtained with AdamW - the additional optimization of the state-free subspace significantly improves performance, resulting in FRUGAL significantly outperforming GaLore (Zhao et al., 2024a). While training with SGD as the state-free optimizer shows somewhat lower performance compared to signSGD, it still significantly outperforms both GaLore and BAdam (Luo et al., 2024). However, we would like to note that unlike signSGD, hyperparameter tuning for SGD training is considerably more challenging. This is because, unlike signSGD, whose update magnitudes approximately equal to those of popular Adam-like algorithms, the magnitude of updates (essentially, gradients) in SGD differs substantially, necessitating learning rates that deviate significantly from those used with the state-full optimizer. Furthermore, successful training with SGD absolutely requires gradient clipping, while the absence of such clipping is not a critical impediment for signSGD.

Table 16: Perplexity of LLaMA-130M models pre-trained on C4 with Lion as state-full optimizer for 200k steps.

| Method | 200k |
|---|---|
| Adam | 18.13 |
| Lion | 18.55 |
| GaLore (+ Lion), $\rho = 0.25$ | 21.65 |
| FRUGAL (+ Lion), $\rho = 0.25$ | **18.89** |

Table 17: Perplexity of LLaMA-130M models pre-trained on C4 for 200k steps with different state-free optimizers for FRUGAL.

| Method | State-free optimizer | Validation perplexity |
|---|---|---|
| Adam | — | 18.13 |
| GaLore, $\rho = 0.25$ | — | 21.11 |
| BAdam, $\rho = 0.25$ | — | 20.34 |
| FRUGAL, $\rho = 0.25$ | signSGD | **18.60** |
| FRUGAL, $\rho = 0.25$ | SGD | 19.11 |

Table 18: Validation perplexity of GPT-2 124M model pre-trained on C4 for 200k steps with various optimization methods and different combinations of sequence length (SL), batch size (BS).

| Method | $\{SL, BS\} = \{256, 512\}$ | $\{SL, BS\} = \{512, 256\}$ |
|---|---|---|
| Adam | 21.94 | 21.90 |
| GaLore, $\rho = 0.25$ | 25.84 | 26.90 |
| BAdam, $\rho = 0.25$ | 25.43 | 26.23 |
| FRUGAL, $\rho = 0.25$ | **23.23** | **23.13** |
| FRUGAL, $\rho = 0.0$ | 25.04 | 24.51 |

**Different architectures.** We have conducted additional experiments on pre-training GPT-2 124M to further strengthen our findings. We followed the setup described in Appendix A.1, except for the tokenizer. We utilized the GPT-2 original tokenizer, with 50257 vocabulary size.

Note, that we have tried two configurations: 1. with sequence length of 256 and batch size of 512 sequences (setup from Zhao et al. (2024a), that we used in our previous experiments), 2. with sequence length of 512 and batch size of 256 sequences (original sequence length of GPT-2).

See results in Table 18. Similarly to experiments with LLaMA, we found that FRUGAL significantly outperforms GaLore and BAdam.

**Computational time.** We present the average computational time of the optimizer step for different sizes of LLaMA models in Table 19. Time is presented in milliseconds. The measurements for memory-efficient methods were made with density $\rho = 0.25$ and update gap $T$ equal to 200. We report the average time over 200 steps (to capture exactly one step with the state-full subspace update). Measurements were conducted on a single A100-80G GPU using PyTorch 2.4.1. We note that these experiments were conducted without using `torch.compile`.

Table 19: Average computational time of optimizer step averaged by 200 steps with update gap 200 for memory-efficient optimizers. We use $\rho = 0.25$ for FRUGAL, Badam and GaLore. Measurements were conducted on a single A100-80G GPU using PyTorch 2.4.1 without `torch.compile`. Time is presented in milliseconds.

| Method | 60M | 130M | 350M | 1B | 3B |
|---|---|---|---|---|---|
| Adam | **3.09** | **6.62** | 17.88 | 62.20 | 124.63 |
| GaLore | 19.50 | 37.06 | 107.11 | 473.72 | 1063.31 |
| BAdam | 39.58 | 29.51 | 63.35 | 71.37 | 123.86 |
| FRUGAL, RandK | 18.16 | 29.65 | 54.94 | 136.55 | 310.11 |
| FRUGAL, Blockwise | 6.70 | 9.76 | **17.49** | **47.49** | **93.49** |

The results show that memory-efficient methods requiring gradient projection within each Linear layer matrix (GaLore, RandK) stand out negatively. GaLore requires more time than RandK due to SVD de-

Table 20: Pre-training LLaMA 3B on C4 dataset for 300K steps. Validation perplexity for different iterations is reported. * indicates runs, that are still in progress.

| Method | 60k | 120k | 180k | 240k | 300k |
|---|---|---|---|---|---|
| Adam | 15.56 | 13.31 | 12.38 | * | * |
| GaLore, $\rho = 0.25$ | 17.37 | 14.94 | * | * | * |
| BAdam, $\rho = 0.25$ | 18.65 | 15.61 | 14.30 | * | * |
| FRUGAL, $\rho = 0.25$ | **15.51** | **13.26** | **12.30** | * | * |
| FRUGAL, $\rho = 0.0$ | 15.68 | 13.39 | * | * | * |
| Training tokens | 6B | 12B | 18B | 24B | 30B |

composition. As model size increases, blockwise-projection methods even start outperforming Adam, despite being implemented through a for-loop over all parameters, while PyTorch uses an efficient Adam implementation by stacking updates into a single shared tensor (flag `foreach=True`) to better utilize the parallelization capabilities of modern GPUs. This occurs because Adam's update step requires significantly more operations than the state-free step in FRUGAL. Therefore, approximately 75% of updates in FRUGAL's for-loop require significantly less time.

**LLaMA 3b experiments.** To evaluate how our method scales to larger model sizes, we conducted pre-training experiments with LLaMA 3B on the C4 dataset. Given the substantial computational costs associated with 3B model experiments, we performed a single run using a uniform learning rate of 1.6e-4 across all methods (learning rate taken from Brown (2020a) Table 2.1), training for 300k steps with gradient clipping set to 1.0 and using a cosine scheduler with 30k warmup steps. Other hyperparameters remain consistent with Appendix A.1. Preliminary results are presented in Table 20.

The results demonstrate that FRUGAL scales excellently to 3B-parameter models, while GaLore and BAdam show significantly inferior performance. Surprisingly, FRUGAL with $\rho = 0.25$ even outperforms Adam. While these results are encouraging, we acknowledge that this performance difference might be attributed to suboptimal hyperparameter selection that potentially favors Linear weights training through signSGD over Adam. For instance, similar to the setup described in Appendix A.1 which we adopted from GaLore, we use a weight decay value of 0.0, which may not be optimal. Despite this caveat, we believe this experiment demonstrates the remarkable potential of FRUGAL for large-scale training.

## G  SIMPLIFIED ALGORITHMS PSEUDOCODE

**Algorithm 4** FRUGAL step pseudocode, PyTorch-like

```
1: def svd_or_randk_step(self):
2:     for param in self.params:
3:         grad = param.grad
4:         param_state = self.state[param]
5:         # update projector if necessary
6:         if self.step % self.update_gap == 0:
7:             param_state["projector"] = self.update_proj(grad)
8:         projector = param_state["projector"]
9:         # obtain state-full grad and state-free grad
10:        grad_full = projector.proj_down(grad)
11:        grad_free = grad_full - projector.proj_up(grad_full)
12:        # reset state-full optimizer state if necessary
13:        if self.step % self.update_gap == 0:
14:            param_state["exp_avg"] = torch.zeros_like(grad_full)
15:            param_state["exp_avg_sq"] = torch.zeros_like(grad_full)
16:        # state-full subspace update
17:        self.step += 1
18:        update_full = self.state_full_step(grad_full, param_state)
19:        update_full = projector.proj_up(update_full)
20:        # state-free subspace update
21:        update_free = self.state_free_step(grad_free)
22:        # perform resulting update
23:        update = update_full + update_free
24:        param.add_(update)
25:
26: def block_step(self):
27:     # change state-full and state-free blocks if necessary
28:     if self.step % self.update_gap == 0:
29:         indices_full = self.update_indices(indices_full)
30:         for idx, param in enumerate(self.params):
31:             grad = param.grad
32:             param_state = self.state[param]
33:             if idx in indices_full:
34:                 # reset state-full optimizer state
35:                 param_state["exp_avg"] = torch.zeros_like(grad)
36:                 param_state["exp_avg_sq"] = torch.zeros_like(grad)
37:                 param_state["full_subspace"] = True
38:             else:
39:                 # free state-full optimizer state to save memory
40:                 param_state.clear()
41:                 param_state["full_subspace"] = False
42:     # perform updates
43:     for param in self.params:
44:         grad = param.grad
45:         param_state = self.state[param]
46:         # choose the optimizer depending on the block type
47:         if param_state["full_subspace"]:
48:             update = self.state_full_step(grad, param_state)
49:         else:
50:             update = self.state_free_step(grad)
51:         # perform resulting update
52:         param.add_(update)
```

**Algorithm 5** Examples of state-full and state-free steps for Algorithm 4

```
1: def state_full_adam_step(self, grad, param_state):
2:     exp_avg = param_state["exp_avg"]
3:     exp_avg_sq = param_state["exp_avg_sq"]
4:     step = self.step
5:     beta1, beta2 = self.betas
6:     exp_avg.mul_(beta1).add_(grad, alpha=1.0-beta1)
7:     exp_avg_sq.mul_(beta2).addcmul_(grad, grad, value=1.0-beta2)
8:     denom = exp_avg_sq.sqrt()
9:     step_size = self.lr_full
10:    if self.correct_bias:
11:        bias_correction1 = 1.0 - beta1 ** step
12:        bias_correction2 = 1.0 - beta2 ** step
13:        step_size = self.lr_full / bias_correction1
14:        bias_correction2_sqrt = math.sqrt(bias_correction2)
15:        denom.div_(bias_correction2_sqrt)
16:    denom.add_(self.eps)
17:    update_full = exp_avg / denom * (-step_size)
18:    return update_full
19:
20: def state_free_signsgd_step(self, grad):
21:     update_free = -self.lr_free * grad.sign()
22:     return update_free
```

## H    LIMITATIONS

We would also like to acknowledge the limitations of this work. Due to computational constraints, we were unable to conduct experiments on pre-training 7B+ LLMs, which is crucial for understanding the potential of our approach when scaling. Furthermore, our experiments are limited to training language models, although memory-efficient optimization could also be beneficial for training diffusion models. Finally, there may be a better method for selecting the next state-full subspace during the training. We leave the exploration of more sophisticated selection strategies for future work.

