# OpenReview forum: "FRUGAL: Memory-Efficient Optimization by Reducing State Overhead for Scalable Training"
_ICLR.cc/2025/Conference — Submitted to ICLR 2025_

### Official Review · Reviewer_yRQJ · 2024-10-31

**Soundness:** 3
**Presentation:** 2
**Contribution:** 2
**Rating:** 5
**Confidence:** 3

**Summary:**

This paper introduces FRUGAL. The fundamental idea is that during the backward pass, we will take a subset of parameters (a block) to perform stateful Adam updates and for the rest parameters (with blockwise selection) or the gradient residuals (with low-rank gradient projection), we use stateless signSGD updates. The memory efficiency of FRUGAL is achieved by reducing the optimizer states. The authors provide a convergence rate similar to SGD momentum's usual rate under nonconvex optimization. The authors also perform experiments with Llama pretraining on C4 and RoBerta-base fine-tuning on GLUE tasks. The baselines are primarily Galore and BAdam.

**Strengths:**

1. FRUGAL's convergence rate is provided and it can recover the rate of standard SGD(M).

2. The experiment execution is strong and the results are convincing. The hyperparameter details are well disclosed and the implementation is provided.

**Weaknesses:**

**Major concern**:

1. The idea of FRUGAL is fairly simple (as a combination of signSGD, Adam, and a gradient projector) but the empirical and theoretical support behind FRUGAL is not solid enough. FRUGAL's stateful optimizer is basically either Galore or BAdam. The main contribution is therefore stateless optimizer part (signSGD), and such effectivenss relies on the finding that stateless optimizers are sufficient to optimize most parameters in LLM (linear weight matrices). The authors only provide a single ablation study in Table 3 without further empirical or theoretical insights on the stateless optimizer part. This evidence alone is not convincing enough on an assured generalization to other non-Llama architectures. So it appears to me that the contribution of this paper is insufficient for an ICLR paper.

2. The motivation (Figure 2) of FRUGAL is that low-rank gradient projection is similar, and random or blockwise selection can cover the whole space. Figure 2 justifies that the top gradient directions across timestep is similar, but *is insufficient to show that random or blockwise selection is always/necessarily better. It is highly likely that after a certain threshold, the role of randomly selected parameters/blocks of parameters have worse performance than top gradient directions. An ablation study on projector type versus stateful optimization density $\rho$ is definitely needed.

**Minor concern**:

1. The presentation of the Algorithm needs to be clearer. It is hard to understand the exact algorithm (which is actually simple) in the first time of reading Algorithm 1 and Section 3.


I consider the first major weakness as critical and I would vote for a borderline reject score at this moment.

**Questions:**

I don't have other questions. All major weaknesses are listed above.

---

> ### Author Response · Authors · 2024-11-20
> **Rebuttal**
>
> We appreciate the reviewer's comprehensive feedback. We are glad that they appreciated the theoretical convergence guarantees, strong experimental results, as well as the provided code implementation.
>
> > The empirical and theoretical support behind FRUGAL is not solid enough
>
> We would like to emphasize that our method consistently beats all baselines, often by a large margin, as demonstrated in Tables 2, 4-7. These results validate the practical effectiveness of FRUGAL across diverse settings.
>
> On the theoretical side, our framework provides the strongest guarantees we are aware of, especially given its ability to handle arbitrary projections. One should note that unlike GaLore, FRUGAL does not require projections to remain fixed for theory (see Theorem 3.8 in GaLore paper).
>
> >FRUGAL's stateful optimizer is basically either Galore or BAdam.
>
> We kindly disagree with the reviewer on this point. FRUGAL does not recover GaLore as a special case, even when the state-free optimizer is discarded. This distinction arises from the reprojection step (Line 7, Algorithm 1), which is critical to enabling FRUGAL to work effectively with arbitrary projections, a capability that sets it apart from GaLore. We provide both discussion and experimental validation of this issue in lines 307-314 and Appendix C.
>
> >This evidence alone is not convincing enough for an assured generalization to other non-Llama architectures.
>
> We have conducted additional experiments on pre-training GPT-2 124M to further strengthen our findings. See details in Table 18. Similarly to experiments with LLaMA, we found that FRUGAL significantly outperforms GaLore and BAdam.
>
> We believe that the current results already provide strong support for both the theoretical and empirical contributions of our work. However, we kindly request the reviewer to specify any additional comparisons, results, or insights that would help address their concerns. We are open to elaborating further or including additional experiments to ensure clarity and completeness.
>
> >An ablation study on projector type versus stateful optimization density $\rho$ is definitely needed.
>
> To address the reviewer's request and further demonstrate the effectiveness of our method, we conducted an additional ablation study. The experiments follow the setup of Table 1 but explore different density $\rho$ values: 0.0625, 0.125, 0.333, and 0.5, while in Table 1 we use $\rho = 0.25$. The results, presented in Table 15, align with our findings from Table 1. Specifically, training with random projection significantly outperforms SVD projection when training without optimizing the state-free subspace. When state-free subspace optimization is employed, SVD projections marginally outperform their Blockwise counterparts.
>
> We believe these empirical results provide sufficient additional evidence to support our conclusions drawn from Figure 2 and Table 1. We remain open to the reviewer's suggestions regarding additional experiments that could further strengthen this argument.
>
> > The presentation of the Algorithm needs to be clearer. It is hard to understand the exact algorithm (which is actually simple) in the first time of reading Algorithm 1 and Section 3.
>
> We appreciate the reviewer's feedback. Indeed, Algorithm 1 is written from a maximally general perspective to cover all possible types of projections. To facilitate understanding of the final algorithm variant used in our experiments, we have provided PyTorch-style pseudocode in Algorithms 4, 5.
>
> To sum up, we believe that we addressed all the reviewer's concerns, none of which is a serious issue with our approach. Furthermore, all the mentioned strengths highlight impactful and timely contributions of our work. Therefore, we would kindly request the reviewer to reconsider their score.

---

> ### Author Response · Authors · 2024-11-25
> **Follow-up to Author Rebuttal**
>
> Dear reviewer yRQJ,
>
> The discussion period is about to end and we would kindly like to hear from you regarding our rebuttal.

---

> ### Comment · Reviewer_yRQJ · 2024-11-30
> **Response to Author's response.**
>
> Thanks for the response.
>
> I previously overlooked Line 7, Algorithm 1 and it is a weakness worthwhile of discussion for Galore. I would suggest add a paragraph of discussion to the main paper besides putting it on Appendix C.
>
> Thanks for the additional experiment on GPT2 and Llama-3B, and the ablation on density $rho$.
>
> I still have concerns on the intuition on most parameters in LLM can be optimized by stateless optimizers part. Even though the authors provide some ablations, **it is better to give an empirical insight beyond the result number (accuracy/perplexity/etc.) itself to strengthen this argument**. The insight means a deeper analysis on any underlying variables or empirical facts that make most parameters in LLM can be optimized effectively by stateless optimizers (something at least can have a causal relationship). I do not see such analysis from this paper so I will still retain my original score.

---

> > ### Author Response · Authors · 2024-12-01
> > **Response to Reviewer's response**
> >
> > We sincerely thank the reviewer for engaging in this discussion with us.
> >
> > > I previously overlooked Line 7, Algorithm 1 and it is a weakness worthwhile of discussion for Galore. I would suggest add a paragraph of discussion to the main paper besides putting it in Appendix C.
> >
> > Thank you for acknowledging this strength of our paper. We already have a paragraph discussing this in the text; please see Lines 307–314. If the reviewer believes this would enhance the presentation of the paper, we are happy to move the content from Appendix C to the main paper for the camera-ready version.
> >
> > > Thanks for the additional experiment on GPT2 and Llama-3B, and the ablation on density $\rho$.
> >
> > Thank you for acknowledging our experiments showing the superiority of FRUGAL when compared to all the relevant baselines.
> >
> > > I still have concerns on the intuition on most parameters in LLM can be optimized by stateless optimizers part. Even though the authors provide some ablations, it is better to give an empirical insight beyond the result number (accuracy/perplexity/etc.) itself to strengthen this argument. The insight means a deeper analysis on any underlying variables or empirical facts that make most parameters in LLM can be optimized effectively by stateless optimizers (something at least can have a causal relationship). I do not see such analysis from this paper so I will still retain my original score.
> >
> > Thank you for this valuable feedback. However, we respectfully disagree that this is the primary contribution of our work. One of the main contributions of our paper is demonstrating that training can be significantly accelerated by updating all parameters in each step while maintaining the memory efficiency of low-rank optimizers, such as Galore. This is a substantial and important contribution to the community, and we already provide a robust empirical evaluation of how to manage the split between state-free and state-full optimizers.
> >
> > We believe that the main reason behind the possibility to optimize most part of the LLM parameters with state-free signSGD lies in the fact that the advantage of the de-facto standard Adam algorithm is precisely that its step resembles a sign update (see discussion in Section 2, lines 145-183). Additional empirical evidence supporting this hypothesis comes from the success of algorithms like Lion [1] and Signum [2], which also take the sign for the final update, although they use momentum buffer, unlike signSGD. The effectiveness of these sign-based methods compared to Adam is well documented e.g. in [3].
> >
> > While gaining a deeper understanding of this phenomenon is an interesting and valuable direction for future research (and likely warrants its own study), we do not believe the absence of such an analysis should justify rejecting the paper. It is common in the ML community for phenomena to be observed first, with detailed understanding emerging later—this does not diminish the contribution of identifying the phenomenon.
> >
> > To sum up, we believe our paper makes significant contributions that are valuable to the community, as acknowledged by the reviewer in other aspects. We would kindly request the reviewer to update their score in light of these considerations.
> >
> > [1] Chen et al. Symbolic Discovery of Optimization Algorithms. Advances in Neural Information Processing Systems, 2023.
> >
> > [2] Balles and Hennig. Dissecting Adam: The Sign, Magnitude and Variance of Stochastic Gradients. International Conference on Machine Learning, 2018.
> >
> > [3] Zhao et al. Deconstructing What Makes a Good Optimizer for Language Models. arXiv:2407.07972, 2024.

---

### Official Review · Reviewer_f3bb · 2024-11-04

**Soundness:** 3
**Presentation:** 2
**Contribution:** 2
**Rating:** 5
**Confidence:** 4

**Summary:**

This work proposed a new memory-efficient training methods that allows part of the parameters being optimization with optimization states within a compact space while other parameters are optimizated in the original space without optimization states. Results on serveral pre-training and fine-tuning tasks demonstrates the effectiveness of the proposed methods.

**Strengths:**

- Plenty of experiments are conducted to evaluate FRUGAL where FRUGAL demonstrates significant improvments against GaLore.

- Both empirical and theoretically justification are provided to validate the effectiveness of FRUGAL.

**Weaknesses:**

- The GLUE benchmarks is little bit outdated, more recent tasks like common-sense reasoning, mt-bench would further improve this work.

- Is there any explanations about which part of the parameters can be directly optimized with SGD type optimizer with other requires adam and why?

- For $\rho=0$ in Table 2, is it equals to fully optimized with SGD? Does it controdict with recent works that demonstrates that transformers can not be effectively optimzied with SGD? [1]

- The concepts of state-full and state-free subspace in line80/82 is hard to understand, it's better to formally define these two concepts.

- line 192: "Surprisingly, we found that although SVD decomposition delivers an initial boost, subsequent training with random projection yields significant improvements", this sequence make it a little bit confusing whether the "Low-rank Random" in Table 1 is training of entire random projection or first SVD and later random.

- it's better to define the meaning of K in the inputs of algorithm 1, as well as s.


[1] Why Transformers Need Adam: A Hessian Perspective

**Questions:**

Please refer to the weakness

---

> ### Author Response · Authors · 2024-11-20
> **Rebuttal**
>
> We would like to thank the reviewer for their detailed comments. We are delighted that they commended the novelty of our extensive experiments and the theoretical and practical justifications of our method.
>
> > The GLUE benchmarks is little bit outdated
>
> Thank you for this feedback. We chose GLUE for experiments because we followed the setup from GaLore (see lines 511-512).
>
> However, we agree that additional experiments on more recent than GLUE fine-tuning benchmarks would provide an additional argument supporting the superiority of our method. We will try to add an additional fine-tuning experiment before the end of the Discussion period.
>
> We would also like to emphasize again that our paper already includes an extensive set of experiments confirming the superiority of our method over baselines, both in pre-training and fine-tuning. Additionally, for this rebuttal, we have already conducted additional experiments on pre-training GPT-2 124M (see details in Table 18), where our method also significantly outperformed the baselines, which once again demonstrates the stability of our method across different setups.
>
> > Is there any explanations about which part of the parameters can be directly optimized with SGD type optimizer with other requires adam and why?
>
> We appreciate the reviewer raising this question. As another reviewer had a similar concern, we have included our response in the general response section.
>
> > For $\rho=0$ in Table 2, is it equal to fully optimized with SGD?
>
> We thank the reviewer for their question. Since a similar question was raised by another reviewer, we have addressed this in the general response.
>
> > The concepts of state-full and state-free subspace in line 80/82 is hard to understand, it's better to formally define these two concepts.
>
> We are grateful for the reviewer's feedback. As it mirrors a question from another reviewer, we have included our response in the general response section.
>
> >this sequence make it a little bit confusing whether the "Low-rank Random" in Table 1 is training of entire random projection or first SVD and later random.
>
> We are very grateful to the reviewer for pointing out this ambiguity. We have reformulated this sentence, see lines 203-205. If the reviewer has any remaining concerns about the clarity, we would be happy to address them.
>
> >it's better to define the meaning of K in the inputs of Algorithm 1, as well as $s$.
>
> We appreciate the reviewer's feedback. We have modified the algorithm description to include $K$'s definition in the Input section and added a brief note clarifying that $s$ denotes the optimizer state. While space constraints prevent us from adding more detailed explanations, we welcome any specific suggestions from the reviewer on how to make the Algorithm 1 more formally precise and clear.
>
> To sum up, we believe that we addressed all the reviewer's concerns, none of which is a serious issue with our approach. Furthermore, all the mentioned strengths highlight impactful and timely contributions of our work. Therefore, we would kindly request the reviewer to reconsider their score.

---

> > ### Author Response · Authors · 2024-11-25
> > **Follow-up to Author Rebuttal**
> >
> > Dear reviewer f3bb,
> >
> > As we approach the end of the discussion period, we would appreciate your feedback on our rebuttal.

---

> > > ### Comment · Reviewer_f3bb · 2024-11-27
> > >
> > > Thanks for the responses. Most of my conerns have been addressed. While I still think finetuning on more recent benchmarks would provide better justification of the proposed methods, Like the setups of a lot of recent works about efficient fine-tuning[1-2].
> > >
> > > [1] S2FT: Efficient, Scalable and Generalizable LLM Fine-tuning by Structured Sparsity.
> > >
> > > [2] OwLore: Outlier-weighed Layerwise Sampled Low-Rank Projection for Memory-Efficient LLM Fine-tuning.

---

> > > > ### Author Response · Authors · 2024-11-29
> > > >
> > > > Thank you for your follow-up and for acknowledging that most of your concerns have been addressed.
> > > >
> > > > We appreciate your suggestion regarding newer fine-tuning benchmarks. While such experiments could provide additional insights, we believe this request should not warrant rejection, given the strong empirical foundation already established in our work. Our evaluations clearly demonstrate the robustness and superiority of our method across diverse setups.
> > > >
> > > > We kindly request that you reconsider your score, as the remaining concern pertains to only additional experiments, while the novelty, theoretical rigor, and strong empirical results already highlight the significant contributions of our work. Given the short timeframe, we will aim to include the requested experiments in the camera-ready version to further strengthen our findings.

---

### Official Review · Reviewer_Cacw · 2024-11-06

**Soundness:** 3
**Presentation:** 3
**Contribution:** 4
**Rating:** 6
**Confidence:** 2

**Summary:**

This paper introduces a novel memory-efficient optimization method. Unlike other state-of-the-art approaches, such as LoRA and GaLore, that have low-rank updates, this method maintains a full-rank update structure. The experimental results demonstrate its superior performance, highlighting its potential advantages in both efficiency and effectiveness over competing methods.

**Strengths:**

**Well-structured Presentation:** The paper is well-structured and easy to follow, with a clear presentation of concepts and methodology.

**Practical Impact:** The method is straightforward to implement and has broad applicability, making it valuable for practical use in various settings.

**Weaknesses:**

**Lack of Discussion on Limitations:** The paper would benefit from a discussion of the method's limitations and potential failure modes. Addressing these aspects would provide a more balanced view of the approach's applicability and constraints.

**Vague Terminology:** Given the importance of "state-full" and "state-free" in the proposed method, the paper should offer clearer definitions of these terms. Precise terminology is essential to fully understand the mechanics and implications of the approach.

**Questions:**

**Formal Definitions of Full and Free States:** Could the authors provide formal definitions of "full" and "free" states as used in the method? A clearer understanding of these terms would improve the paper’s theoretical foundation.

**Main Limitations:** What are the primary limitations of this approach? A discussion on the constraints or situations where the method might be less effective would help clarify its scope and potential trade-offs.

**Running Time Comparisons:** Beyond memory efficiency, how does the method’s running time compare to that of other baseline approaches? Performance in terms of speed is crucial for practical deployment, so direct comparisons would provide a more complete picture of the method’s efficiency.

---

> ### Author Response · Authors · 2024-11-20
> **Rebuttal**
>
> We thank the reviewer for their detailed review. We are glad that they liked our paper's presentation and our method's practical impact.
>
> >The paper would benefit from a discussion of the method's limitations and potential failure modes.
>
> We thank the reviewer for this suggestion. To provide a more comprehensive understanding of potential trade-offs associated with our method, we have added a Limitations section to Appendix H.
>
> >Given the importance of "state-full" and "state-free" in the proposed method, the paper should offer clearer definitions of these terms.
>
> We thank the reviewer for this feedback. Since this overlaps with another reviewer's inquiry, we have addressed it in the general response.
>
> > Beyond memory efficiency, how does the method’s running time compare to that of other baseline approaches?
>
> As requested by the reviewer, we have measured the running time of all methods used in the paper. We present the average computational time of the optimizer step for different sizes of LLaMA models in Appendix F, Table 19. The measurements for memory-efficient methods were made with density $\rho=0.25$ and update gap $T$ equal to $200$. We report the average time over $200$ steps (to capture precisely one step with the state-full subspace update). Measurements were conducted on a single A100-80G GPU using PyTorch 2.4.1. We note that these experiments were conducted without using torch.compile.
>
> The results show that memory-efficient methods requiring gradient projection within each Linear layer matrix (GaLore, RandK) stand out negatively. GaLore requires more time than RandK due to SVD decomposition. As model size increases, blockwise-projection methods even start outperforming Adam, despite being implemented through a for-loop over all parameters, while PyTorch uses an efficient Adam implementation by stacking updates into a single shared tensor (flag `foreach=True`) to better utilize the parallelization capabilities of modern GPUs. This occurs because Adam's update step requires significantly more operations than the state-free step in FRUGAL. Therefore, approximately 75% of updates in FRUGAL's for-loop require significantly fewer computations and, consequently, less time.
>
> To sum up, we believe that we addressed all the reviewer's concerns, none of which is a serious issue with our approach. Furthermore, the reviewer highlighted the practicality of our approach. Therefore, we would kindly request the reviewer to reconsider their score.

---

> > ### Author Response · Authors · 2024-11-25
> > **Follow-up to Author Rebuttal**
> >
> > Dear reviewer Cacw,
> >
> > With the discussion deadline approaching, we would appreciate your thoughts on our rebuttal response.

---

> > > ### Comment · Reviewer_Cacw · 2024-11-26
> > >
> > > Thank you for your response. I have reviewed the opinions of other reviewers. My concern is whether the empirical results are comprehensive enough to justify publishing a relatively simple idea in this venue. However, given that my initial evaluation is already positive and that I am not an expert in this field, I have decided to retain my score.

---

> > > > ### Author Response · Authors · 2024-11-30
> > > >
> > > > Thank you for retaining your positive evaluation.
> > > >
> > > > We would like to encourage the reviewer to also consider our responses to the other reviewers, as they provide additional clarifications and context that may be beneficial.
> > > >
> > > > Regarding the simplicity of our method, we believe that its simplicity is a strength rather than a limitation. While the approach may seem straightforward in hindsight, it is novel and required a fresh perspective to identify and develop. To the best of our knowledge, no prior work has proposed a similar methodology, and we provide both strong empirical evidence and theoretical justification to support its efficacy and relevance.
> > > >
> > > > Additionally, simplicity often facilitates practical adoption, making our method more accessible for real-world applications, which we believe aligns well with the goals of this venue.
> > > >
> > > > We hope this perspective helps address your concern, and we remain available to provide further clarifications if needed.

---

### Official Review · Reviewer_YtQB · 2024-11-07

**Soundness:** 3
**Presentation:** 1
**Contribution:** 3
**Rating:** 5
**Confidence:** 3

**Summary:**

This paper introduces FRUGAL (Full-Rank Updates with GrAdient spLitting) that reduces memory consumption by splitting gradient updates into two subspaces. A *state-full* subspace is updated using advanced optimization algorithms like Adam, while a *state-free* subspace is updated using stateless and memory-efficient methods like SGD or signSGD. The framework allows for a flexible choice of optimizers and projection methods. FRUGAL achieves state-of-the-art results in pre-training and fine-tuning tasks, outperforming existing memory-efficient algorithms while maintaining a similar memory budget.

**Strengths:**

-1) The paper presents a novel approach to improving memory efficiency while performing updates using full-rank information.
-2) The proposed method is flexible, supporting various choices for both stateful and stateless optimizers as well as different projection methods. It offers convergence guarantees for FRUGAL within a specified framework and consistently outperforms existing memory-efficient algorithms, such as GaLore and BAdam, achieving performance levels close to the memory-intensive Adam optimizer.
-3) Additionally, the paper provides valuable insights into the learning dynamics of transformer models.

**Weaknesses:**

-1) The paper's structure would greatly benefit from a clearer organization. Currently, some analysis and experimental results appear within the Methods section, which disrupts the logical flow and makes it challenging for readers to follow the methodology. Reorganizing the paper and dedicating specific sections to distinct aspects of the research could significantly enhance readability and impact.

-2) Several notations (e.g., g~) are introduced without proper definitions, which assumes too much prior knowledge from readers. Additionally, concepts like smoothness and unbiasedness are only vaguely referenced and would benefit from clearer definitions. The theory section should be expanded to explicitly define each notation and assumption, as well as to contextualize them within a more general setting relevant to the proposed method.

-3) Including a full-parameter fine-tuning baseline in Table 4 would provide a valuable benchmark, offering a clearer context for evaluating the results.

-4) Definitions for Full-Rank SVD/Random and Low-Rank SVD/Random are scattered across Table 1 and lack clear differentiation. Consolidating these explanations into a concise paragraph would improve clarity and reader comprehension.

-5) Lastly, there are deviations from the primary algorithm, such as using column-wise projection instead of block-wise projection. For completeness, it would be beneficial to include results using the original proposed approach alongside the variations in the experiments.

-6) By solving this issues in the revision, especially following a more structured writing style and lowering the jumps, the paper would definitely level up.

**Questions:**

- 1) Including more experiments comparing the method with various stateful and stateless optimizers would enhance the paper.

- 2) Testing models with larger sizes (e.g., 3B and 7B) could further demonstrate the generalizability of the proposed method.

- 3) Please clarify the reasons for selecting the specific optimizers in the theoretical section. They appear restrictive and differ from those used in the main algorithm. Additional details and guarantees would help generalize this proof.

- 4) While it’s mentioned that stateless optimizers typically underperform with transformer architectures, the paper doesn’t explain why FRUGAL with $\rho=0$ achieves optimal performance in certain scenarios. Providing more details and comparisons would clarify this.
Expanding the dataset and incorporating diverse architectures could strengthen the argument for FRUGAL's superior characteristics.

---

> ### Author Response · Authors · 2024-11-20
> **Rebuttal: Part 1**
>
> We are grateful to the reviewer for their thorough review. We appreciate that they commended the novelty of our idea, the theoretical and practical justifications of our method, as well as our results in analyzing the training dynamics of transformer models.
>
> >The paper's structure would greatly benefit from a clearer organization. Currently, some analysis and experimental results appear within the Methods section, which disrupts the logical flow and makes it challenging for readers to follow the methodology.
>
> We appreciate the reviewer's feedback. To enhance the paper's readability and clarity, we have reorganized the Method section. Specifically, we have divided it into two sections: 1. Empirical Analysis and Motivation, which includes subsections ''The importance of exploring the entire space during the training process'' and ''Advantage of the Full-Rank Updates''. 2. FRUGAL: Full-Rank Updates with GrAdient spLitting, which includes the subsection ''FRUGAL as Optimization Framework''. We have also restructured the text within this section: to achieve a more coherent narrative flow, we moved the introduction of the general framework with reference to Algorithm 1 to the beginning of the section.
>
> However, we believe that the presence of experiments later in this section remains appropriate. As FRUGAL is positioned as a general framework that supports arbitrary types of projections, State-Full and State-Free optimizers, these experiments are crucial for justifying our specific implementation choices. While future research may propose improved hyperparameter settings and components that could be incorporated into FRUGAL, we needed to identify and validate the optimal configuration from existing options for our main experimental evaluation. Therefore, maintaining these experiments within this section provides the necessary justification for our implementation choices and serves as a foundation for potential future improvements to the framework.
>
> If the reviewer has additional suggestions regarding the structure of this section, we welcome their input and would be happy to incorporate them further.
>
> > Several notations (e.g., g~) are introduced without proper definitions, which assumes too much prior knowledge from readers. Additionally, concepts like smoothness and unbiasedness are only vaguely referenced and would benefit from clearer definitions.
>
> We thank the reviewer for highlighting these opportunities for improvement. In response, we have incorporated definitions for $\tilde g$ and random sample $\zeta$, as well as a definition for the L-smooth function.
>
> However, we have maintained the current formulation of Unbiasedness as presented in Assumption 1, as it is both precise and consistent with standard definitions in the literature (e.g., [1]).
>
> We welcome any further suggestions from the reviewer regarding potential improvements to the theory section.
>
> > Including a full-parameter fine-tuning baseline in Table 4 would provide a valuable benchmark
>
> We thank the reviewer for this suggestion. We have added the full-parameter fine-tuning baseline to Table 4. We would like to note that, following [2], rather than conducting these experiments ourselves, we have adopted the results from prior works.
>
> >Definitions for Full-Rank SVD/Random and Low-Rank SVD/Random are scattered across Table 1 and lack clear differentiation.
>
> To improve the readability of Table 1, we have modified the column headers. Instead of a single 'Method' column, we have added two columns: 'Projection type' and 'Optimizes state-free subspace', which better reflect the distinctions between the algorithms compared in this table. Additionally, for consistency with the text, we have added clarifications in lines 206-208, 250-251, and 296-298.
>
> If the reviewer believes this presentation can be further improved, we welcome their feedback.
>
> > Lastly, there are deviations from the primary algorithm, such as using column-wise projection instead of block-wise projection.
>
> We appreciate the reviewer's attention to this discrepancy. To ensure complete coherence throughout the paper, we have revised the text at the end of this subsection accordingly. Please refer to lines 302-306.
>
> We welcome any additional suggestions from the reviewer regarding potential improvements to this section of the paper.

---

> ### Author Response · Authors · 2024-11-20
> **Rebuttal: Part 2**
>
> > Including more experiments comparing the method with various stateful and stateless optimizers would enhance the paper.
>
> Following the reviewer's request, we conducted additional experiments. We explored two variations: 1) replacing AdamW with Lion as the state-full optimizer, and 2) substituting signSGD with SGD as the state-free optimizer.
>
> The results are presented in Tables 16 and 17. As observed, the results for Lion are comparable to those obtained with AdamW - the additional optimization of the state-free subspace significantly improves performance. Training with SGD as the state-free optimizer also demonstrates satisfactory results. However, we would like to note that unlike signSGD, hyperparameter tuning for SGD training is considerably more challenging. This is because, unlike signSGD, whose update magnitudes approximately equal to those of popular Adam-like algorithms, the magnitude of updates (essentially, gradients) in SGD differs substantially, necessitating learning rates that deviate significantly from those used with the state-full optimizer. Furthermore, successful training with SGD absolutely requires gradient clipping, while the absence of such clipping is not a critical impediment for signSGD.
>
> > Testing models with larger sizes (e.g., 3B and 7B) could further demonstrate the generalizability of the proposed method.
>
> We agree with the reviewer on this matter. The paper already includes preliminary results for training LLaMA 7b in Appendix D, Table 14. While these results don't allow for a fair comparison with standard Adam training (because 1. the GaLore baseline uses 8-bit Adam, which apparently significantly degrades the final result, and 2. the training was conducted in pure bfloat-16 format), we believe, as stated in lines 991-992, that these results demonstrate FRUGAL's potential for scaling.
>
> Furthermore, to more thoroughly examine our method's potential with increasing model sizes, we are working on experiments comparing Adam and FRUGAL training on a 3B model. Since this experiment requires substantial computational resources, obtaining results takes considerable time. However, we plan to share preliminary results within a week before the end of the Discussion period and include the final results in the camera-ready version of the paper.
>
> > Please clarify the reasons for selecting the specific optimizers in the theoretical section.
>
> We appreciate the reviewer’s comment on the selection of specific optimizers in the theoretical section. The primary reason for choosing SGD with momentum (SGDM) and SGD in our theoretical analysis is the lack of guaranteed convergence for Adam in certain scenarios, as demonstrated by  Reddi et al. (2018) [4]. Therefore, we discard normalization in our analysis, leading to a combination of SGDM and SGD.
>
> Additionally, we note that methods such as GaLore focus solely on gradient descent (according to Theorem 3.8 in GaLore paper, analysis is limited to constant projection of updates). In contrast, FRUGAL's framework is designed to accommodate arbitrary projections, significantly broadening its applicability and demonstrating its theoretical superiority over existing methods. This flexibility underpins the superiority of our algorithmic choices.
>
> > The paper doesn’t explain why FRUGAL with $\rho = 0$ achieves optimal performance in certain scenarios.
>
> W thank the reviewer for raising this point. As it mirrors a question from another reviewer, we have included our detailed response in the general response.
>
> > Incorporating diverse architectures could strengthen the argument for FRUGAL's superior characteristics
>
> We have conducted additional experiments on pre-training GPT-2 124M to further strengthen our findings. See details in Table 18. Similarly to experiments with LLaMA, we found that FRUGAL significantly outperforms GaLore and BAdam.
>
> To sum up, we believe that we addressed all the reviewers' concerns, none of which were serious issues, mainly focusing on issues with presentation. Furthermore, all the mentioned strengths highlight impactful and timely contributions of our work. Therefore, we would kindly request the reviewer to reconsider their score.
>
> [1] Liu et. al., An improved analysis of stochastic gradient descent with
> momentum. NeurIPS 2020.
>
> [2] Hu et. al., Lora: Low-rank adaptation of large language models. ICLR 2022.
>
> [3] Zhao et. el., Deconstructing what makes a good optimizer for language models. arXiv:2407.07972, 2024.
>
> [4] Reddi et al. On the convergence of adam and beyond. arXiv preprint arXiv:1904.09237, 2019.

---

> > ### Author Response · Authors · 2024-11-25
> > **Follow-up to Author Rebuttal**
> >
> > Dear Reviewer YtQB,
> >
> > As promised, we have initiated experiments on LLaMA 3B pre-training using the C4 dataset and would like to share our current findings. As of now, all the five methods have completed at least 40% of their training, and we have included these intermediate results in Table 20, with the experimental setup and conclusions detailed in Appendix F of the revision. In brief, the results demonstrate that FRUGAL scales very well to 3B-parameter models, while GaLore and BAdam show significantly inferior performance.
> >
> > In addition, before the discussion period closes, we would be very grateful for your feedback on our earlier response.

---

> > > ### Comment · Reviewer_YtQB · 2024-11-27
> > > **Thank you for your response**
> > >
> > > I appreciate the authors’ detailed responses. However, I have several concerns:
> > >
> > > 1. A key aspect of the proposed method is its strategy for determining which layers or parameters should be optimized using a stateful optimizer versus a state-free one. While Table 3 presents some results, it lacks an analysis of the selection of linear layers.
> > >
> > > 2. How do you propose bridging the gap between theoretical proofs and practical implementation in the context of optimizer selection?
> > >
> > > 3. In Table 15, SVD consistently outperforms other selection methods, albeit with potentially significant computational costs. This seems inconsistent with the paper’s claim that random projection offers superior performance. Clarification on this discrepancy is necessary.
> > >
> > > 4. Density studies, being a core aspect of the algorithm, should be included in the main text. A more organized and thorough investigation would improve the paper’s presentation.
> > >
> > > 5. Although the Lion optimizer is included in the study, its performance is not compared with that of Adam in any table. Additionally, the paper does not explore the combinations of these optimizers with GaLore or the proposed method.
> > >
> > > 6. The experiments are presented in a scattered manner, making it difficult to follow them and draw clear conclusions.
> > >
> > > Considering these points, I retain my previous scores.

---

> ### Author Response · Authors · 2024-11-29
> **Follow up Part 1/2**
>
> We thank the reviewer for the response. We address their concern in detail below.
>
> >A key aspect of the proposed method is its strategy for determining which layers or parameters should be optimized using a stateful optimizer versus a state-free one. While Table 3 presents some results, it lacks an analysis of the selection of linear layers.
>
> We respectfully disagree with the statement that this is a key aspect of our work. While our paper presents multiple contributions, the mentioned aspect is not central to our findings. The primary contributions are the reprojection of states and the updating of all network parameters.
> The reviewer's comment seems to address the lack of explanation regarding why we initially considered only Linear layers as candidates for the state-free subspace in all experiments except those presented in Table 3 and Table 13.
> To address this, we followed the setup from GaLore, where the authors also projected gradients onto a low-rank subspace solely for Linear layers. This approach is natural, given our primary objective: minimizing memory consumption. As model sizes increase, Linear layers in Transformers account for a growing proportion of the total parameters. For example, in LLaMA 3B, Linear layers constitute approximately 94% of all model parameters. Consequently, the memory demands of other layer types are comparatively negligible, which justifies focusing on Linear layers in this context.
> We have included this clarification in the paper on line 194 and footnote 2. We welcome additional feedback if the reviewer has further questions or if we have misunderstood their concern.
>
> >How do you propose bridging the gap between theoretical proofs and practical implementation in the context of optimizer selection?
>
> We would like to reiterate that our theory provides **the strongest justification** compared to related work. As we argue, our theory is **the closest to what is actually implemented** when compared to related work. However, as discussed, the difference arises from fundamental challenges in analyzing adaptive algorithms such as Adam.
>
> Additionally, our theory cannot be used to determine which parameters should be selected for a state-free optimizer versus a stateful optimizer. However, this limitation is not specific to our theory, as SGD is already the worst-case optimal algorithm. Consequently, we cannot achieve better results when considering combinations of SGDM and SGD. Nevertheless, our results are tight, as we recover the best-known outcomes for both SGD and SGDM under the analyzed setting.
>
> >In Table 15, SVD consistently outperforms other selection methods, albeit with potentially significant computational costs. This seems inconsistent with the paper’s claim that random projection offers superior performance. Clarification on this discrepancy is necessary.
>
> There is no inconsistency here. We never claim that random projection offers superior performance **in general**. On the contrary, we assert that Random projection outperforms SVD when training **without optimizing state-free subspace** (see lines 200-208). Furthermore, in Table 1 and lines 296-301, we explicitly state that "SVD outperforms both RandK and Block projections" when training **with optimizing state-free subspace**. In the same paragraph, we explain our ultimate choice of Blockwise projection over SVD based on reduced computational and memory costs while maintaining comparable (just slightly worse) performance.
>
> The results in Table 15 are fully aligned with our previous findings in Table 1 and our statements in lines 200-208 and lines 296-301. While SVD significantly underperforms compared to Random projection when training **without optimizing state-free subspace** (which further supports our hypothesis from Section 3.1), SVD shows slightly better results than Blockwise projection when training **with optimizing state-free subspace**.
>
> We note that we compared explicitly against Blockwise projection rather than Random projection because they showed similar results in Table 1, leading us to focus on Blockwise projection in subsequent experiments as part of FRUGAL. We included these entries for completeness and to maintain consistency with Table 1's style.

---

> > ### Author Response · Authors · 2024-11-29
> > **Follow up Part 2/2**
> >
> > >Density studies, being a core aspect of the algorithm, should be included in the main text. A more organized and thorough investigation would improve the paper’s presentation.
> >
> > We respectfully disagree with the reviewer's assertion that any of the density studies in the Appendix (specifically, Table 11 and Table 15) are crucial for understanding our framework's potential. We believe that the results presented in Table 2 for FRUGAL with $\rho=0.0$ and $\rho=0.25,$ and for Adam (which is, essentially, FRUGAL with $\rho=1.0$) provide sufficient insight into how little the performance degrades as $\rho$ decreases from $1.0$ to $0.0$. The results in Table 11 were included solely for completeness. The main purpose of Table 15 is to demonstrate that the hypothesis from Section 3.1 (which pertains only to the Empirical analysis for training **without optimizing state-free subspace** and it has no direct relation to the final FRUGAL method) holds true across different density values $\rho$.
> >
> > If the reviewer has specific suggestions regarding additional investigations that could be conducted concerning density studies, we would welcome their input.
> >
> > >Although the Lion optimizer is included in the study, its performance is not compared with that of Adam in any table.
> >
> > We would like to emphasize that the objective of our research is not to identify **the best optimizer in general,** but rather to **develop a memory-efficient optimization framework**. Like previous works in this direction (GaLore, BAdam), we propose an enhancement that reduces the memory budget for existing optimization algorithms that require large memory for their states.
> >
> > Therefore, our focus is not on comparing different optimizers (such as Lion and Adam) with each other but rather on examining the relationship between the results of an **original optimizer** (e.g., Adam or Lion) and **various memory-efficient approaches built on top of that optimizer** (specifically, GaLore, BAdam, and FRUGAL). Consequently, we do not consider the performance comparison between Adam and Lion critical. However, for easier interpretation of the results, we have added an entry with Adam's results to Table 16.
> >
> > >Additionally, the paper does not explore the combinations of these optimizers with GaLore or the proposed method.
> >
> > We believe there might be a misunderstanding: in Table 16 and Appendix F, we specifically present a comparison between the original Lion optimizer, its combination with GaLore (entry "GaLore (+ Lion), $\rho=0.25$"), and its combination with our proposed method (entry "FRUGAL (+ Lion), $\rho=0.25$"). Furthermore, the combinations of GaLore and FRUGAL with Adam are, in fact, the methods labeled as "GaLore" and "FRUGAL" in all other tables throughout the paper (Table 2, Tables 4-7, etc.).
> >
> > >The experiments are presented in a scattered manner, making it difficult to follow them and draw clear conclusions.
> >
> > We respectfully disagree with the reviewer's assessment. We believe that all main results in the main paper (Tables 1-4) are presented sequentially and described in detail. Tables 5-14 in the Appendix serve either as ablation studies for specific hyperparameters of the method or as verification of FRUGAL's robustness under varying external training hyperparameters. Since these tables don't contain any unexpected results, we believe the current format of presenting these results is sufficient for completeness. Tables 15 and beyond were added during the rebuttal phase, which explains their separate location from other tables (we did it for the reviewers' convenience); we plan to adjust their placement in the camera-ready version.
> >
> > However, if the reviewer has *specific* suggestions on how we could improve the presentation of experiments in our paper, we would greatly appreciate their input.

---

### Official Review · Reviewer_2Mhk · 2024-11-09

**Soundness:** 2
**Presentation:** 2
**Contribution:** 3
**Rating:** 5
**Confidence:** 4

**Summary:**

The work proposes a memory efficient training method called FRUGAL which is essentially a combination of full-rank updates with gradient splitting. The authors partition the parameters and update using advanced optimizers (like Adam) for low-dimensional updates and state-free methods (like SGD or signSGD) for remaining directions. Additionally, the authors provide theoretical convergence guarantees and validate FRUGAL’s effectiveness through experiments on models like LLaMA.

**Strengths:**

1. The combination of state-free optimizers with advanced ones, like SGD and Adam, for memory efficient training is a novel idea.
2. The  empirical results show that FRUGAL does better than other methods in terms of memory use and perplexity,
3. The paper includes sufficient ablation studies and it helps to see how FRUGAL works in different situations and settings.

**Weaknesses:**

Line 249 introduces state-free and stateful parameters but could provide more explicit explanation on the selection criteria. Are parameters randomly selected to each category? In that case the assumption is all the parameters are equally important for that iteration. The work could benefit from more detailed study on how to choose the parameters for state free updates.

The purpose of the density parameter ($\rho$) is not thoroughly explained, especially in relation to zero-density training. Please clarify whether zero-density training implies all parameters are state-free (i.e., trained exclusively with SGD). The selection of $\rho$ is not mentioned in the algorithm as well.

**Questions:**

GaLore theoretically prove that gradient is low-rank and a study in BlockLLM (https://arxiv.org/pdf/2406.17296)  show that only a few parameters are updated during the training. A few other recent works also seem to suggest that the low rank structure exists in the network. But this paper seems to suggest the opposite. Do you see a space where these two ideas coexist? For example, low rank for certain tasks vs full rank for other tasks?

Minor:
- Introduce abbreviations for better readability. For example SGD as Stochastic Gradient Descent.
- Missing references Adam-mini and BlockLLM

---

> ### Author Response · Authors · 2024-11-20
> **Rebuttal: Part 1**
>
> We thank the reviewer for their valuable feedback. We are pleased that they appreciated the novelty of our idea, our empirical results, and extensive ablation studies.
>
> >Line 249 introduces state-free and stateful parameters but could provide more explicit explanation on the selection criteria. Are parameters randomly selected to each category? In that case the assumption is all the parameters are equally important for that iteration. The work could benefit from more detailed study on how to choose the parameters for state free updates.
>
>  We thank the reviewer for their insightful comment on the selection criteria for state-free and stateful parameters. Our primary contribution emphasizes updating all parameters, which is crucial for achieving state-of-the-art performance, as detailed in Sections 3.1 and 3.2. This approach contrasts with methods such as GaLore, which only updates a subset of parameters during each iteration.
>
> In our work, we extensively explore various parameter selection strategies, as highlighted in Table 1 and the discussion in Sections 3.1 and 3.2. Among these, we consider both random parameter selection approaches (RandK, Random projection) and deterministic ones (SVD decomposition, blockwise projection with descending/ascending block changes). While a comprehensive study on the importance of individual parameters may be valuable, it is beyond the scope of this work. Such an analysis would require developing additional adaptive metrics for parameter importance assessment, and we leave this investigation for future work.
>
> Nonetheless, we believe that our investigations in Sections 3-4 and Table 1, which examine 4 different types of state-free subspace selection, cover a sufficient range of possible projections. Furthermore, in Table 3, we conduct an additional study regarding the relative importance of different parameters. If the reviewer has specific suggestions regarding any other particular algorithms for state-free subspace selection, we would be happy to consider them and incorporate them into our paper.
>
> > The purpose of the density parameter (ρ) is not thoroughly explained, especially in relation to zero-density training. Please clarify whether zero-density training implies all parameters are state-free (i.e., trained exclusively with SGD). The selection of ρ is not mentioned in the algorithm as well.
>
> Thank you for this question. Given that another reviewer raised a similar point, we have provided our answer in the general response.
>
> > But this paper seems to suggest the opposite. Do you see a space where these two ideas coexist? For example, low rank for certain tasks vs full rank for other tasks?
>
> We thank the reviewer for this excellent question. Indeed, GaLore theoretically proves that the gradient is low-rank, and we confirmed this finding in our initial experiments - for a 60M model with hidden size of 512, the gradients stable ranks varied between 1 and 10, depending on the layer.
>
> However, we believe our work does not contradict the low-rank nature of gradients. Our results show that although the gradient's Frobenius norm is predominantly concentrated in the leading singular values, utilizing the remaining part of the gradient is crucial for successful training. We propose the following potential explanation for this phenomenon: gradient magnitudes are generally proportional to weight magnitudes. However, even weights with small values can be critical for the overall LLM performance, as the neural network function is non-linear, meaning small weight changes can lead to significant output variations. Furthermore, while individual updates may be low-rank, the resulting weight matrix after training typically becomes high-rank, as demonstrated in Chen et al., 2021 (Figure 2).
>
> Chen, Patrick, et al. "Drone: Data-aware low-rank compression for large nlp models." Advances in neural information processing systems 34 (2021): 29321-29334.
>
> >Introduce abbreviations for better readability. For example SGD as Stochastic Gradient Descent.
>
> We are grateful to the reviewer for this comment. Indeed, we introduce SGD in the abstract without expanding the abbreviation. However, this was done to avoid overloading the abstract. When SGD is first mentioned in the main text of the paper, we provide its full form. We also believe that specifically in the case of SGD, using this abbreviation in the abstract without expansion is appropriate because Stochastic Gradient Descent is widely known and considered general knowledge.
>
> If the reviewer can point out other instances in the text where we use abbreviations without proper expansion, we would be grateful for such feedback.

---

> > ### Author Response · Authors · 2024-11-20
> > **Rebuttal: Part 2**
> >
> > > Missing references Adam-mini and BlockLLM
> >
> > We are grateful to the reviewer for this suggestion. While Adam-mini is already mentioned in the paper (see citation on line 051 and reference in lines 131-132), Block-LLM is indeed a recent relevant work that we overlooked. We have added a citation to this work on line 078. Despite addressing similar problems, a meaningful comparison with Block-LLM, unfortunately, is not feasible as their code is not published, and comparing against their reported results would be methodologically incorrect as they, like GaLore, use pure bfloat16 training, whereas our evaluation focuses on Mixed-Precision training (see discussion in lines 408-419). If the reviewer has any additional suggestions, we would be happy to answer them.
> >
> >
> > To sum up, we believe that we addressed all the reviewer's concerns, none of which is a serious issue with our approach. Furthermore, all the mentioned strengths highlight impactful and timely contributions of our work. Therefore, we would kindly request the reviewer to reconsider their score.

---

> > ### Comment · Reviewer_2Mhk · 2024-11-25
> > **Thank the authors for their response**
> >
> > I thank the authors for their detailed response to my feedback.  I am retaining my previous score.

---

> ### Author Response · Authors · 2024-11-25
>
> We thank the reviewer for acknowledging our response. However, as we have provided detailed explanations addressing each concern, we would appreciate clarification on what specific issues remain that justify maintaining the original score. We would be happy to further engage in discussion if there are some unresolved concerns.

---

> > ### Comment · Reviewer_2Mhk · 2024-11-29
> > **Response feedback**
> >
> > My main concern is that randomly choosing parameters does not perform as well as SVD.  But performing SVD is computationally expensive. More discussion on this could be useful.

---

> > > ### Author Response · Authors · 2024-11-30
> > >
> > > We kindly ask the reviewer to clarify and elaborate on the formulation of their concern. We are not quite clear how the experimentally confirmed trade-off between performance versus computational and memory requirements of our framework's two different configurations could be considered **a weakness of our paper and grounds for rejection.**
> > >
> > > We also kindly request specific guidance regarding the desired direction for discussion, as we believe the paper already contains a substantial discussion of this trade-off's results (Table 1, lines 289-301, Appendix B).

---

### Author Response · Authors · 2024-11-20
**General Response: Part 1**

We would like to thank all the reviewers for their time spent reviewing our manuscript and for their valuable feedback.

Before proceeding to the main part of our general response, we would like to draw reviewers' attention to an important change we made after the submission. We discovered a minor bug in the evaluation code for pre-training experiments in the GaLore codebase, which we used as a foundation for our experiments. This bug was related to an incorrect value of `total_batches` in the `evaluate_model` function of the `torchrun_main.py` file. As a result, the final evaluation perplexity values were slightly lower than they should have been.

We have recalculated all results and updated the values in the revised version of the paper. We would like to specifically emphasize that this bug only affects the absolute values of perplexity but does not impact the relative ranking of methods - that is, if method A performed better than method B previously, it remains superior under the correct perplexity calculations.

Furthermore, we are grateful that the reviewers appreciated several key strengths of our work:

1. Novelty: The combination of stateful and stateless optimizers for memory-efficient training while maintaining full-rank information was highlighted as an innovative approach (Reviewers 2Mhk, YtQB).
2. Theoretical foundations: The paper provides convergence guarantees, with FRUGAL able to recover the rate of standard SGDM (Reviewers YtQB, f3bb, yRQJ).
3. Practical value: FRUGAL is straightforward to implement and demonstrates broad applicability across various settings (Reviewer Cacw).
4. Strong experimental validation and extensive ablation: Our comprehensive experiments and ablation studies demonstrate FRUGAL's superior performance compared to existing methods like GaLore, with well-documented implementation details and hyperparameters (Reviewers 2Mhk, YtQB, f3bb, yRQJ).
5. Reproducibility: We have provided code for all the conducted experiments and supported it with detailed description of the experimental setup (Reviewer yRQJ).
6. Technical insights: The work provides valuable understanding of transformer learning dynamics (Reviewer YtQB).

We have also enhanced the paper with additional experiments and incorporated the revisions suggested by the reviewers:

### Experiments:
* Added additional pre-training experiments with non-LLaMA architecture (GPT-2 124M), as requested by Reviewer yRQJ and suggested by Reviewer YtQB
* Added an ablation study on projector type versus density $\rho$, as requested by Reviewer yRQJ
* Included additional experiments with other state-full and state-less optimizers, as requested by Reviewer YtQB
* Added full-parameter fine-tuning baseline for Table 4, as requested by Reviewer YtQB
* Conducted comparison of optimizer step runtime, as requested by Reviewer Cacw


### Text Revisions
In addition, we have improved our presentation as requested by the reviewers.
#### Major changes
* Provided more detailed description of the density parameter $\rho$ and the meaning of zero-density training, which were questioned by Reviewers 2Mhk and f3bb
* Reorganized the method section by splitting it into two parts: Empirical Analysis and Method, as suggested by Reviewer YtQB
* Added simplified PyTorch-like pseudocode of Algorithm 1 for a more specific case, as advised by Reviewer yRQJ
* Added Limitations section (Appendix H) that the Cacw reviewer requested
* Included formal definitions of several notations and definitions, as requested by Reviewer YtQB
* Modified method names in Table 1 to improve readability, as noted by Reviewer YtQB

#### Minor changes
* Fixed minor inconsistencies in the algorithm description, which were pointed out by Reviewer YtQB
* Rephrased ambiguous sentences, as noted by Reviewer f3bb
* Clarified Algorithm 1, as requested by Reviewer f3bb
* Added citation to the relevant BlockLLM paper, as suggested by Reviewer 2Mhk

All the changes in the main paper are highlighted with **blue** color. Newly added experiments, text, and Algorithms are located in Appendix F-H.

---

> ### Author Response · Authors · 2024-11-20
> **General Response: Part 2**
>
> ### General Inquiries
>
> Also, multiple reviewers raised similar questions, so we addressed them in the general response.
>
> >Reviewers Cacw and f3bb asked to provide a formal definitions of **state-full** and **state-free**.
>
> To enhance the readability and clarity of our paper, we have added more formal definitions of **state-full** and **state-free** subspaces and optimizers in the Introduction (see paragraphs in lines 077-090). If the reviewers believe these definitions could be further improved, we are open to suggestions.
>
> > Reviewers YtQB and f3bb were curious about the explanation of which certain parts of the parameters can be optimized with a state-free optimizer and why.
>
> We thank the reviewers for raising this interesting question. In Table 3, Table 13, and lines 471-481 and 526-532, we provide detailed experimental investigation and discussion of which model modules are sensitive to the choice between Adam and other less advanced optimization algorithms.
>
> As we write in lines 480-481, our results show the remarkable sensitivity of Logits to optimizer choice, which is aligned with findings from [1]. The authors of that paper hypothesized that this sensitivity is related to different columns of the Logits matrix corresponding to different tokens that appear with dramatically different frequencies in texts, and it is precisely here that the adaptive per-parameter learning rate enabled by Adam becomes crucial. However, we want to specifically note that this explanation remains a hypothesis and requires separate investigation to fully understand this phenomenon (for instance, similar token-related logic applies to the Embeddings matrix, yet it doesn't exhibit comparable sensitivity). Such investigation is an interesting direction for future research but lies outside the scope of our current work.
>
> We would also like to emphasize that, to the best of our knowledge, our paper is the first to demonstrate that almost all modules in the model can be optimized using state-free optimizers. We believe this contribution already represents a significant advancement for further research into this phenomenon.
>
> > Reviewers 2Mhk and f3bb raised questions about the definition of $\rho$, particularly about the meaning of zero-density training with $\rho=0$, and how it differs from training all parameters using a state-free optimizer.
>
> We would like to note that this parameter was introduced in lines 456-457 (previous revision) with reference to Appendix A.1, where it was described in lines 774-776. We also emphasize that training with $\rho=0$ is not equivalent to training the entire model with a state-free optimizer but rather indicates that only all Linear layers are trained using the state-free optimizer, while the remaining parameters are trained using standard Adam (this setup is similar to GaLore). This is also explicitly noted in the caption of Table 2 and in the paragraph dedicated to zero-density training in lines 471-481. Moreover, lines 471-481 and Table 3 are devoted to understanding why training the entire model with a state-free optimizer yields such poor quality, while training with $\rho=0$ does not degrade performance significantly.
>
> To make the description of the density parameter $\rho$ clearer and more comprehensible, we have made several clarifications to the paper. Specifically, we expanded the introduction of parameter $\rho$ in lines 460-463, added its description in the caption of Table 2, included its mention for context in line 473, and extended the explanation of how $\rho$ relates to rank $r$ from GaLore in Appendix A.1 line 774-778.
>
> We welcome any questions from the reviewers regarding the value of this parameter or suggestions about its description in the paper.
>
> [1] Zhao et. el., Deconstructing what makes a good optimizer for language models. arXiv:2407.07972, 2024.

---

### Meta-Review · Area_Chair_y4U3 · 2024-12-23

**Metareview:**

This paper proposes FRUGAL, a memory-efficient optimization framework that aims to provide full-rank updates for large language model (LLM) training by splitting parameters into those updated with stateful optimizers (e.g., Adam) and those updated with state-free methods (e.g., signSGD). The authors offer theoretical convergence guarantees and present empirical evaluations that, at face value, suggest the method outperforms existing approaches like GaLore and BAdam under fixed memory budgets.


Strengths:
	•	The paper addresses a timely and important problem: memory-efficient optimization for large-scale LLM training.
	•	The authors combine adaptive (Adam-like) and sign-based updates, exploring a “split” that preserves a full-rank update signal.
	•	Comprehensive experiments and extensive ablations demonstrate performance gains over some baselines.
	•	Theoretical analysis is provided, aiming to justify the algorithmic design and giving some convergence guarantees.

**Additional Comments On Reviewer Discussion:**

Concerns and Reasons for Rejection:
	1.	Empirical Justification and Generality:
While the authors show improvements on certain architectures (notably LLaMA and GPT-2 124M) and tasks (pre-training and GLUE fine-tuning), multiple reviewers pointed out the lack of broader evidence to convincingly generalize beyond these settings. The key claim—most parameters can be trained effectively with a state-free optimizer—remains insufficiently justified. The paper shows that signSGD works surprisingly well for most parameters, but provides no deeper insight or causal explanation for why this should be the case. Without stronger intuition, broader architectural coverage, or theoretical insight specifically addressing this phenomenon, the foundational claim lacks the rigor needed for a strong contribution.
	2.	Comparison and Clarity on Method Variations:
The method’s presentation, especially during the discussion phase, raised concerns about scattered experiments and the clarity of how certain projections and densities are selected. While the authors added experiments and clarifications, reviewers still found them insufficiently systematic and somewhat difficult to follow. Key aspects—such as why certain subsets of parameters are chosen for stateful vs. state-free updates—remain largely heuristic and not convincingly resolved.
	3.	Intuition and Underlying Mechanisms Absent:
A central premise of FRUGAL is that large portions of the parameter space do not need sophisticated adaptive updates, yet the paper provides no substantial underlying analysis or intuition beyond numerical improvement. Several reviewers requested deeper insight into what distinguishes the parameters or subspaces that can be effectively handled by signSGD from those that cannot, but the authors did not address this in a sufficiently illuminating manner. Lack of this deeper understanding was repeatedly cited as a weakness that reduces confidence in the paper’s longer-term significance.
	4.	Simplicity and Unanswered Reviewer Concerns:
While the idea is conceptually simple, simplicity by itself is not a drawback if accompanied by thorough validation and deep insight. However, the reviewers felt that the results, though promising, did not fully substantiate the strong claims or provide enough rigorous evidence to elevate the approach beyond incremental improvement. Several reviewers maintained their stance that further experiments, broader evaluations, or more compelling explanations are necessary for acceptance.

Conclusion:
Though the paper makes an effort to address memory-efficiency and presents interesting initial findings, the reviewers concluded that it does not convincingly demonstrate why the proposed approach is broadly applicable, nor does it provide sufficient explanatory depth. Multiple reviewers maintained their initial reservations even after the author responses. Given the persisting concerns, the paper does not currently meet the bar for acceptance at ICLR.

---

### Decision · Program_Chairs · 2025-01-22

Reject